# SmoothMix: Training Confidence-calibrated Smoothed Classifiers for Certified Robustness

**Jongheon Jeong**[1]     **Sejun Park**[2][*]     **Minkyu Kim**[3]

**Heung-Chang Lee**[4]     **Doguk Kim**[5][†]     **Jinwoo Shin**[3,1]

[1]School of Electrical Engineering, KAIST     [2]Vector Institute for Artificial Intelligence
[3]Kim Jaechul Graduate School of AI, KAIST     [4]Kakao Enterprise
[5]Department of Artificial Intelligence, Inha University
{jongheonj, minkyu.kim, jinwoos}@kaist.ac.kr
sejun.park@vectorinstitute.ai
andrew.com@kakaoenterprise.com   dgkim@inha.ac.kr

## Abstract

*Randomized smoothing* is currently a state-of-the-art method to construct a *certifiably robust* classifier from neural networks against $\ell_2$-adversarial perturbations. Under the paradigm, the robustness of a classifier is aligned with the *prediction confidence*, *i.e.*, the higher confidence from a smoothed classifier implies the better robustness. This motivates us to rethink the fundamental trade-off between accuracy and robustness in terms of *calibrating* confidences of a smoothed classifier. In this paper, we propose a simple training scheme, coined *SmoothMix*, to control the robustness of smoothed classifiers via *self-mixup*: it trains on convex combinations of samples along the direction of adversarial perturbation for each input. The proposed procedure effectively identifies over-confident, near off-class samples as a cause of limited robustness in case of smoothed classifiers, and offers an intuitive way to adaptively set a new decision boundary between these samples for better robustness. Our experimental results demonstrate that the proposed method can significantly improve the certified $\ell_2$-robustness of smoothed classifiers compared to existing state-of-the-art robust training methods.[3]

## 1 Introduction

*Adversarial examples* [53, 17] in deep neural networks clearly highlight that neural networks often generalize differently from humans, at least without an additional prior of *local smoothness* of predictions with respect to the input space: an adversarially-crafted, yet imperceptible input perturbation can drastically change the prediction of a neural network based classifier. Due to the intrinsic complexity of neural networks, however, the community has noticed that it is extremely hard to directly encode this smoothness prior into neural networks [6, 3, 55], especially without relying on *adversarial training* [40, 66], *i.e.*, augmenting training data with its adversarial examples. Even with adversarial training, (a) the non-convex, minimax nature of the training introduces many optimization difficulties, often resulting in a harsh over-fitting [51, 47], and (b) it is generally hard to provably guarantee that the learned classifier is indeed smooth.

---

[*]Work done at KAIST.

[†]Work done at Kakao Enterprise.

[3]Code is available at https://github.com/jh-jeong/smoothmix.

35th Conference on Neural Information Processing Systems (NeurIPS 2021).

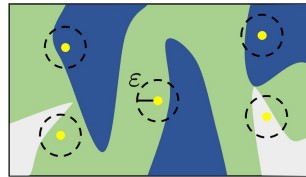 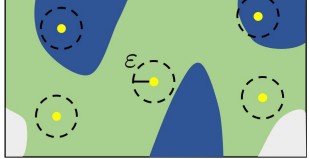 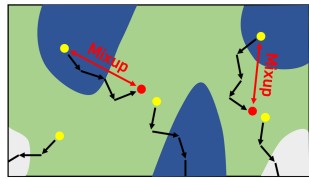

| (a) Adversarial training [40] | (b) SmoothAdv [49] | (c) SmoothMix (Ours) |

Figure 1: Illustrations of how each training method obtains adversarial robustness: adversarial training [40] considers an $\varepsilon$-ball around each sample and corrects adversarial examples found in these balls; SmoothAdv [49] directly employs adversarial training on smoothed classifiers; SmoothMix (ours) can be differentiated from SmoothAdv as it (*i*) does not assume an explicit norm restriction on adversarial examples, and (*ii*) applies *mixup* [67] instead of correcting the adversarial examples.

*Randomized smoothing* [33, 11] is relatively a recent idea that aims to *indirectly* encode the smoothness prior: Cohen et al. [11] have shown that any classifier, regardless of whether it is smooth or not, can be transformed into a *certifiably robust* classifier via averaging its predictions over Gaussian noise, where the (certified) robustness of the classifier depends on how well the base classifier performs with the noise. Compared to adversarial training, this notion of "indirect" smoothness can be favorable in a sense that (a) it is easier to optimize, and (b) offers a provable guarantee on the robustness. Currently, randomized smoothing is considered as the state-of-the-art approach in the context of building a neural network based classifier that is certifiably robust on $\ell_2$-perturbations [39].

In this respect, a growing body of research has focused on improving the robustness guarantee that randomized smoothing can give, *e.g.*, via different smoothing measures [34, 62] or improved certification procedures [14, 43]. One of important directions on this line of research is to investigate *which training* of the base classifier could maximize the certified robustness of smoothed classifiers [49, 65, 23]. In particular, Salman et al. [49] have proposed *SmoothAdv*, showing that employing adversarial training [40] for smoothed classifiers could further improve the robustness, akin to the standard neural networks. This motivates us to develop a new form of adversarial training, more specialized for smoothed classifiers.

**Contribution.** In this paper, we propose *SmoothMix*, a novel adversarial training method designed for improving the certified robustness of smoothed classifiers. One of the key features that smoothed classifiers offer is a direct correspondence from *prediction confidence* to adversarial robustness: achieving a higher confidence in a smoothed classifier implies that the classifier can give a better certified robustness. Inspired by this, we found that the certified robustness of a given data sample can be significantly decreased by nearby *off-class* but *over-confident* [45] inputs: such "harmful" inputs would occupy an unnecessarily large robust radius near the sample of our interest, especially when they do not contain much information to be discriminated by the classifier.

Under the finding, we aim to calibrate the confidence of these off-class inputs to improve the certified robustness at the original input. More specifically, we first observe that such over-confident examples can be efficiently found along the direction of *adversarial* perturbations for a given input. Then, we suggest to regularize the over-confident predictions along the adversarial direction toward the *uniform* prediction through a *mixup* loss [67] (see Figure 1 for an overview). This new approach of incorporating adversarial examples effectively permits more distant examples in training, even when they go off-class, based on the local-smoothness of smoothed classifiers. It also suggests an intuitive way of defining confidence beyond the given data samples to smoothed classifiers.

We evaluate our proposed SmoothMix against with various state-of-the-art robust training methods for smoothed classifiers on a wide range of image classification benchmarks, including MNIST [32], CIFAR-10 [29], and ImageNet [48] datasets. Overall, the results consistently show that our new adversarial training scheme for smoothed classifiers significantly improves the certified robustness compared to existing methods, *e.g.*, one of our CIFAR-10 model could largely outperform an existing state-of-the-art result on the average certified robustness in $\ell_2$-radius $0.720 \rightarrow 0.737$. Through an extensive ablation study, we also verify that our method is (a) robust to the choice of hyperparameters, and (b) can effectively trade-off between the accuracy and robustness [66] of smoothed classifiers.

Overall, our work suggests that the robustness of a classifier should be set individually per sample considering its nearby inputs: we approach this problem by leveraging the relationship between the

prediction confidence and robustness of smoothed classifiers. Recently, there have been also some initial attempts to incorporate a *sample-wise* treatment for robustness by allowing input-dependent noise scales in randomized smoothing [1, 58, 10]. However, our theoretical analysis shows that such an approach would eventually suffer from the curse of dimensionality (Theorem 1 in Appendix B), highlighting our approach of focusing on a "better calibration" as a promising alternative.

## 2  Preliminaries

We assume an *i.i.d.* dataset $\mathcal{D} = \{(x_i, y_i)\}_{i=1}^n \sim P$, where $x_i \in \mathbb{R}^d$ and $y_i \in \mathcal{Y} := \{1, \cdots, C\}$, and focus on the problem of correctly classifying a given input $x$ into one of $C$ classes. Let $f : \mathbb{R}^d \to \mathcal{Y}$ be a classifier modeled by $f(x) := \arg\max_{c \in \mathcal{Y}} F_c(x)$ with $F : \mathbb{R}^d \to \Delta^{C-1}$, where $\Delta^{C-1}$ denotes the probability simplex in $\mathbb{R}^C$. For example, $F$ can be a neural network followed by a softmax layer.

In the context of *adversarial robustness*, we require $f$ not only to correctly classify $(x, y) \sim P$, but also to be *locally-constant* around $x$, *i.e.*, $f$ should not contain any adversarial examples around $x$. In this respect, one can measure and attempt to maximize the adversarial robustness of a classifier $f$ by considering the *minimum-distance* of adversarial perturbation [44, 7, 8], namely:

$$R(f; x, y) := \min_{f(x') \neq y} \|x' - x\|_2. \tag{1}$$

**Randomized smoothing.**  In cases when $f$ is too complex to control its predictions in practice, *e.g.*, if $f$ is a neural network on high-dimensional data, directly solving and maximizing (1) can be hard. *Randomized smoothing* [11] instead constructs a new classifier $\hat{f}$ from $f$ that is easier to obtain robustness by transforming the base classifier $f$ with a certain *smoothing measure*, where in this paper we focus on the case of Gaussian distributions $\mathcal{N}(0, \sigma^2 I)$:

$$\hat{f}(x) := \arg\max_{c \in \mathcal{Y}} \mathbb{P}_{\delta \sim \mathcal{N}(0, \sigma^2 I)} \left( f(x + \delta) = c \right). \tag{2}$$

For a given $(x, y)$, Cohen et al. [11] have shown that $R(\hat{f}; x, y)$ can be lower-bounded by the *certified radius* $\underline{R}(\hat{f}, x, y)$, which can be derived from the *confidence* of $\hat{f}$ at $x$, namely we denote it by $p_f(x)$:

$$R(\hat{f}; x, y) \geq \sigma \cdot \Phi^{-1}(p_f(x)) =: \underline{R}(\hat{f}, x, y) \text{ where } p_f(x) := \mathbb{P}_{\delta \sim \mathcal{N}(0, \sigma^2 I)}(f(x + \delta) = \hat{f}(x)), \tag{3}$$

provided that $\hat{f}(x) = y$, and otherwise $R(\hat{f}; x, y) := 0$.[4] This lower bound is known as tight for the $\ell_2$-minimum distance, *e.g.*, the bound is optimal for linear classifiers [11].

Although randomized smoothing can be applied for any classifier $f : \mathbb{R}^d \to \mathcal{Y}$, the robustness of smoothed classifiers can vary depending on $p_f$ as in (3), *i.e.*, how $f$ performs on a given input under the presence of Gaussian noise. In this sense, to obtain a robust $\hat{f}$, Cohen et al. [11] simply propose to train $f$ using Gaussian augmentation by default:

$$\min_F \mathbb{E}_{\substack{(x, y) \sim P \\ \delta \sim \mathcal{N}(0, \sigma^2 I)}} \left[ \mathcal{L}(F(x + \delta), y) \right], \tag{4}$$

where $\mathcal{L}$ denotes the standard cross-entropy loss.

**Adversarial training for smoothed classifiers.**  To obtain $f$ that gives a more robust classifier when smoothed into $\hat{f}$, Salman et al. [49] propose *SmoothAdv* that employs adversarial training [40] on $\hat{f}$:

$$\min_{\hat{f}} \max_{\|x' - x\|_2 \leq \epsilon} \mathcal{L}(\hat{f}; x', y). \tag{5}$$

Due to the intractability of $\hat{f}$, however, it is hard to directly optimize the inner maximization of (5) via gradient methods. To bypass this, SmoothAdv attacks the *soft-smoothed* classifier $\hat{F} := \mathbb{E}_\delta[F_y(x + \delta)]$ instead. Specifically, SmoothAdv finds an adversarial example via solving the following:

$$\hat{x} = \arg\max_{\|x' - x\|_2 \leq \epsilon} \mathcal{L}(\hat{F}; x', y) \approx \arg\max_{\|x' - x\|_2 \leq \epsilon} \left( -\log\left( \frac{1}{m} \sum_i F_y(x' + \delta_i) \right) \right), \tag{6}$$

using Monte Carlo integration with $m$ samples of $\delta$, namely $\delta_1, \cdots, \delta_m \sim \mathcal{N}(0, \sigma^2 I)$.

---

[4]Here, $\Phi$ denotes the cumulative distribution function of the standard normal distribution.

# 3  Method

Our goal in this paper is to develop a more suitable form of adversarial training (AT) for smoothed classifiers, taking into account their unique characteristics on adversarial robustness over standard neural networks. Figure 1 illustrates a motivating example: as shown in Figure 1(a), AT typically assumes a fixed-sized ball of radius $\varepsilon$ that each adversarial perturbation must be in, as the goal of the training is to defend the classifier against adversaries under a specific threat model. However, in a case when AT is applied to a smoothed classifier, *e.g.*, as done by SmoothAdv, this assumption may be too restrictive, particularly for inputs where the classifier already certifies robustness of radii larger than $\varepsilon$ (*e.g.*, Figure 1(b)). This demands for a new form of AT specially for smoothed classifiers, *e.g.*, that allows more distant adversarial examples, despite its fundamental difficulty in the context of standard neural networks [25, 69].

In this regard, our proposed training method of *SmoothMix* takes a completely different approach to incorporate adversarial examples during training. More specifically, for a given sample $(x, y) \sim P$, our method finds an adversarial example of $x$ *without* an explicit norm constraint, *i.e.*, "unrestrictively." This is because our focus is not to find an input for correcting its label to $y$ (as in the standard AT), but to find an input that is *over-confident* and *semantically off-class*, *i.e.*, it is not beneficial to the classifier to label this input to $y$. Once we have such an example, SmoothMix then labels it as the *uniform confidence*, and considers a *mixup* training [67] with the original $x$: by linearly interpolating with the uniform confidence, SmoothMix effectively calibrates the over-confident inputs in between, re-balancing the certified radius at the original sample of $x$ at the end.

## 3.1  Exploring over-confident adversarial examples in smoothed classifiers

Recall that we have a (base) classifier $f$ of the form $f(x) = \arg\max_{c \in \mathcal{Y}} F_c(x)$, $\hat{f}$ is its smoothed counterpart, and we aim to improve the robustness of $\hat{f}$ by incorporating adversarial examples in training. In this paper, we are particularly interested in adversarial examples of $\hat{f}$ that is found *without* a hard restriction in its perturbation size. More concretely, for a given training sample $(x, y) \sim P$, we find adversarial examples by solving the following optimization:

$$\tilde{x} := \arg\max_{x'} \left( \mathcal{L}(\hat{f}; x', y) - \beta \cdot \|x' - x\|_2^2 \right), \tag{7}$$

where $\mathcal{L}$ is the cross-entropy loss, and $\beta > 0$ is to ensure that (7) cannot be arbitrarily far from $x$.

As proposed by Salman et al. [49] (see Section 2), one can optimize (7) by approximating the intractable $\hat{f}$ with the soft-smoothed classifier $\hat{F} := \mathbb{E}_\delta[F(x + \delta)]$, in a similar manner to (6). Based on this approximation, we simply perform a $T$-step gradient ascent from $\tilde{x}^{(0)} := x$ with step size $\alpha > 0$ to solve (7) using $m$ samples of $\delta$, namely $\delta_1, \cdots, \delta_m \sim \mathcal{N}(0, \sigma^2 I)$:[5]

$$\tilde{x}^{(t+1)} := \tilde{x}^{(t)} + \alpha \cdot \frac{\nabla_x J(\tilde{x}^{(t)})}{\|\nabla_x J(\tilde{x}^{(t)})\|_2}, \text{ where } J(x) := -\log\left( \frac{1}{m} \sum_i F_y(x + \delta_i) \right). \tag{8}$$

Figure 2 demonstrates two particular instances of these "unrestricted" adversarial examples found from (8) on $x$, and plots how the confidence of inputs changes as they are linearly interpolated from the clean input to its adversarial counterpart. From this illustration, we make several remarks those would lead to a more direct motivation to our method:

- We observe that adversarial perturbations found via (8), *i.e.*, from a smoothed classifier, could contain enough amount of semantic changes even in a perceptual sense, in either ways of translating the input to another class (Figure 2(a)), or simply removing some relevant information for the current class (Figure 2(b)). At least for these cases, therefore, it is reasonable for the classifier to keep its low confidence to the original class. Such a "perceptually-aligned" representation is not a unique property of smoothed classifiers, but has been generally observed on adversarially-robust classifiers [50, 15, 26]: in other words, we leverage the provable robustness of smoothed classifiers during training to reasonably obtain a semantically off-class samples, those will be labeled as the uniform confidence.

---

[5]Here, we note that the $\beta$-term in (7) are omitted in (8). In practice, we do not use nor tune $\beta$ in our method mainly for simplicity, as the role of $\beta$ can be replaced by assuming a finite $\alpha \cdot T$, *i.e.*, by the *Lagrangian duality*: an unconstrained optimization with $\ell_2$-regularization implicitly defines a hard constraint in its $\ell_2$-norm.

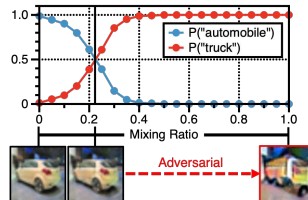 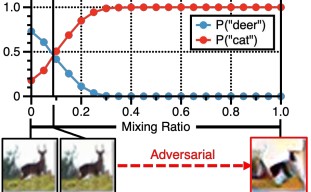 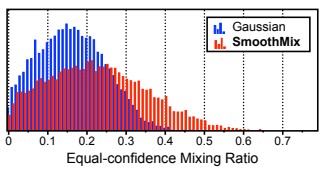

(a) In-class translation       (b) Out-of-class translation

Figure 2: Illustration of adversarial examples unrestrictively found in CIFAR-10 with a smoothed ResNet-110 ($\sigma = 0.25$). The plot demonstrates the change of confidence between two classes as the input is linearly interpolated.

Figure 3: *Equal-confidence mixing ratios* on CIFAR-10, *i.e.*, the minimal mixing ratios for changing the correct prediction when each input is linearly interpolated to its adversarial example.

Table 1: Comparison of the average (a) true-class confidence, and (b) maximum off-class confidence of smoothed classifiers, for adversarial inputs searched via PGD from CIFAR-10 test samples around $\ell_2$-ball of radius $\varepsilon$. We use ResNet-110 trained on CIFAR-10 with $\sigma = 0.5$ for this comparison.

| CIFAR-10 (Test set; %) | Clean | $\varepsilon = 1.0$ | $\varepsilon = 2.0$ | $\varepsilon = 3.0$ | $\varepsilon = 4.0$ | $\varepsilon = 5.0$ |
|---|---|---|---|---|---|---|
| (a) $\mathbb{E}[\mathbb{P}(f(x + \delta) = y)]$ | **66.4** | 47.1 | 24.3 | 14.2 | 11.3 | 10.7 |
| (b) $\mathbb{E}[\max_{c \neq y} \mathbb{P}(f(x + \delta) = c)]$ | 24.2 | 37.8 | 59.5 | **71.8** | **78.5** | **82.0** |

- A major problem we rather highlight here is the tendency of *over-confidence* [45] toward the direction of adversarial perturbation: Figure 2 also presents how the confidence of the given smoothed classifier changes as we linearly interpolate the input from $x$ to $\tilde{x}$. Overall, the adversarially-crafted samples $\tilde{x}$ usually attain significantly higher confidence compared to that of $x$, consequently their *certified radius* (3) would be much larger as well. Therefore, considering that $\tilde{x}$ are still nearby $x$, such the over-confidence at $\tilde{x}$ would negatively affect the certified radius of $x$, especially when $\tilde{x}$ does not contain much semantically meaningful information as observed in Figure 2(b).[6] This observation is further supported qualitatively in Table 1: by comparing the average (a) *true-class* confidence and their (b) *off-class* confidences of a smoothed classifier, those are defined by (a) $\mathbb{E}[\mathbb{P}(f(x + \delta) = y)]$ and (b) $\mathbb{E}[\max_{c \neq y} \mathbb{P}(f(x + \delta) = c)]$, respectively, we confirm that the off-class confidence of $\tilde{x}$ can be abnormally higher than those of clean samples $x$ as we allow more budget on $\varepsilon$.

### 3.2 SmoothMix for confidence-calibrated training of smoothed classifiers

Based on the observations from Section 3.1, we hypothesize that the *miscalibration* of confidences between $x$ and its unrestricted adversarial example $\tilde{x}$ is an important factor that degrades the certified robustness of smoothed classifiers, and propose to penalize the over-confidence by mixing the *uniform* confidence to them. More concretely, we consider the *mixup* [67] training between $x$ and $\tilde{x}$, *i.e.*, by augmenting the given training data with the following pairs:

$$x^{\texttt{mix}} := (1 - \lambda) \cdot x + \lambda \cdot \tilde{x}^{(T)}, y^{\texttt{mix}} := (1 - \lambda) \cdot \hat{F}(x) + \lambda \cdot \frac{\mathbb{1}}{C}, \text{ where } \lambda \sim \mathcal{U}\left(\left[0, \tfrac{1}{2}\right]\right) \quad (9)$$

where $\hat{F}(x) \in \Delta^{C-1}$ is the soft-smoothed prediction of $x$, $\mathcal{U}$ denotes the uniform distribution, $\lambda$ is a random variable that represents the mixing ratio between $(x, \hat{F}(x))$ and $(\tilde{x}^{(T)}, \mathcal{U}(\mathcal{Y}))$, and $\mathbb{1}$ denotes the $C$-dimensional vector of ones. Here, we notice that $\lambda$ is sampled only from $[0, \frac{1}{2}]$, unlike the standard choice [67] of $\mathcal{U}([0, 1])$: recall from Figure 2(a) that $\tilde{x}$ can be often semantically in-class, so that a direct supervision of the uniform confidence on it could harm the classifier. By simply taking only the half part of the mixed samples closer to $x$, we could reasonably avoid these cases while maintaining its effect to prevent the over-confidence issue. The actual loss to minimize for these new data simply follows the cross-entropy loss with Gaussian augmentation, similarly to (4):

$$L^{\texttt{mix}} := \mathbb{E}_{\delta \sim \mathcal{N}(0, \sigma^2 I)} \left[\mathcal{L}(F(x^{\texttt{mix}} + \delta), y^{\texttt{mix}})\right]. \quad (10)$$

---

[6]We nevertheless remark that such $\tilde{x}$ is still sufficiently far from $x$ compared to the adversarial examples commonly used in the standard adversarial training (and SmoothAdv), that has a hard $\ell_2$-norm restriction.

Recall the over-confidence issue observed in Figure 2 as we follow from $x$ to $\tilde{x}^{(T)}$. Minimizing $L^{\mathtt{mix}}$ (10) directly corresponds to calibrating the high confidence at $\tilde{x}^{(T)}$ and the samples in-between, while keeping the original prediction of $\hat{F}(x)$ at $x$. Even in cases that $\tilde{x}^{(T)}$ does not have an over-confidence issue, *i.e.*, when its prediction is already close to the uniform confidence, the loss (10) would assign a relatively low value for $\tilde{x}^{(T)}$ so that it can act only if there exists an overconfident $\tilde{x}^{(T)}$ nearby $x$.

**Incorporating SmoothAdv for free.**   As our method focuses on adversarial examples that are moderately far from the original inputs assuming that the classifier is already locally-smooth, one may still enjoy the effectiveness of SmoothAdv if it could further enforce the local smoothness. We indeed observe that the joint training can be helpful for the robustness of smoothed classifiers, but a naïve combination of them could incur too much costs for finding separate adversarial examples for each method. Instead, we found that simply taking $x \leftarrow \tilde{x}^{(1)}$ without modifying our current training, *i.e.*, using the *single-step adversarial example* found during (8) instead of the clean sample, can reasonably bring a similar effect. In this respect, we allow SmoothMix to use $(\tilde{x}^{(1)}, y)$ instead of $(x, y)$ depending on demand of more robustness at expense of decreased clean accuracy.

**Overall training.**   Combining the proposed loss with the standard Gaussian training (4) gives the full objective to minimize for our training method. For a given sample $(x, y) \sim P$, and by letting $L^{\mathtt{nat}} := \mathbb{E}_\delta \left[ \mathcal{L}(F(x + \delta), y) \right]$, the final loss of SmoothMix is given by:

$$L := L^{\mathtt{nat}} + \eta \cdot L^{\mathtt{mix}} \tag{11}$$

where $\eta > 0$ is a hyperparameter to control the trade-off between accuracy and robustness. Algorithm 1 in Appendix A demonstrates a concrete training procedure of SmoothMix using $m$ samples of $\delta$ for the Monte Carlo approximation.

## 4   Experiments

We evaluate the effectiveness of our method extensively on MNIST [32], CIFAR-10 [29], and ImageNet [48][7] classification datasets. Overall, the results consistently highlight that our newly proposed training can significantly improve the certified robustness of smoothed classifiers compared to existing robust training methods. We point out the improvements are especially remarkable on the certified accuracy at larger perturbations, at which SmoothMix mainly focus compared to prior arts. We also conduct an ablation study on the proposed method to convey a detailed analysis on the individual components. The detailed experimental setups, *e.g.*, training details, datasets, and hyperparameters for the baseline methods, are specified in Appendix C.

**Baseline methods.**   We compare our method with a variety of existing techniques proposed for a robust training of smoothed classifiers, as listed in what follows: (a) Gaussian [11]: standard training with Gaussian augmentation; (b) Stability training [38]: a cross-entropy regularization between $F(x)$ and $F(x + \delta)$; (c) SmoothAdv [49]: adversarial training on smoothed classifier; (d) MACER [65]: a regularization that maximizes an approximative form of the certified radius (3); and (e) Consistency [23]: a KL-divergence based regularization that minimizes the variance of $F(x + \delta)$ across $\delta$. Whenever possible, we use the pre-trained models released by authors for our evaluation to reproduce the baselines. The more detailed training configurations are specified in Appendix C.2.

**Evaluation metrics.**   Our evaluation of the robustness for a given smoothed classifier $\hat{f}$ is largely based on the protocol proposed by Cohen et al. [11], similarly to prior works [49, 65, 23]: more concretely, Cohen et al. [11] proposed a practical Monte Carlo based certification procedure, namely CERTIFY, that returns the prediction of $\hat{f}$ and a "safe" lower bound of certified radius over the randomness of $n$ samples with probability at least $1 - \alpha$, or abstains the certification.

From CERTIFY, we consider two evaluation metrics: (a) the *approximate certified test accuracy* at various radii: the fraction of the test dataset which CERTIFY classifies correctly with radius larger than $r$ without abstaining, and (b) the *average certified radius* (ACR) [65]: the average of certified radii returned by CERTIFY on the test dataset counting only the correctly classified samples, namely $\mathrm{ACR} := \frac{1}{|\mathcal{D}_{\mathtt{test}}|} \sum_{(x,y) \in \mathcal{D}_{\mathtt{test}}} \mathrm{CR}(f, \sigma, x) \cdot \mathbf{1}_{\hat{f}(x)=y}$, where $\mathcal{D}_{\mathtt{test}}$ is the test dataset, and $\mathrm{CR}$ denotes the certified radius from $\mathrm{CERTIFY}(f, \sigma, x)$. Here, the latter metric, ACR, is for a better comparison of robustness under trade-off between accuracy and robustness [56, 66]: by its definition, ACR naturally

---

[7]Results on the ImageNet dataset can be found in Appendix G.

Table 2: Comparison of approximate certified test accuracy (%) and ACR on MNIST. All the models are trained and evaluated with the same smoothing factor specified by $\sigma$. Each value except ACR indicates the fraction of test samples which have $\ell_2$ certified radius larger than the threshold specified at the top row. We set our results bold-faced whenever the value improves the Gaussian baseline, and underlined whenever the value improves the best among the considered baselines.

| $\sigma$ | Models (MNIST) | ACR | 0.00 | 0.25 | 0.50 | 0.75 | 1.00 | 1.25 | 1.50 | 1.75 | 2.00 | 2.25 | 2.50 | 2.75 |
|---|---|---|---|---|---|---|---|---|---|---|---|---|---|---|
| | Gaussian [11] | 0.911 | 99.2 | 98.5 | 96.7 | 93.3 | 0.0 | 0.0 | 0.0 | 0.0 | 0.0 | 0.0 | 0.0 | 0.0 |
| | Stability training [38] | 0.915 | 99.3 | 98.6 | 97.1 | 93.8 | 0.0 | 0.0 | 0.0 | 0.0 | 0.0 | 0.0 | 0.0 | 0.0 |
| | SmoothAdv [49] | 0.932 | 99.4 | 99.0 | 98.2 | 96.8 | 0.0 | 0.0 | 0.0 | 0.0 | 0.0 | 0.0 | 0.0 | 0.0 |
| | MACER [65] | 0.920 | 99.3 | 98.7 | 97.5 | 94.8 | 0.0 | 0.0 | 0.0 | 0.0 | 0.0 | 0.0 | 0.0 | 0.0 |
| 0.25 | Consistency [23] | 0.928 | 99.5 | 98.9 | 98.0 | 96.0 | 0.0 | 0.0 | 0.0 | 0.0 | 0.0 | 0.0 | 0.0 | 0.0 |
| | **SmoothMix** ($\eta = 1.0$) | **0.931** | **99.5** | **98.9** | **98.2** | **96.4** | 0.0 | 0.0 | 0.0 | 0.0 | 0.0 | 0.0 | 0.0 | 0.0 |
| | **+ One-step adversary** | **0.933** | **99.4** | **99.0** | **98.2** | **96.9** | 0.0 | 0.0 | 0.0 | 0.0 | 0.0 | 0.0 | 0.0 | 0.0 |
| | **SmoothMix** ($\eta = 5.0$) | **0.932** | **99.4** | **99.0** | **98.2** | **96.7** | 0.0 | 0.0 | 0.0 | 0.0 | 0.0 | 0.0 | 0.0 | 0.0 |
| | **+ One-step adversary** | **0.933** | **99.3** | **99.0** | **98.2** | **97.0** | 0.0 | 0.0 | 0.0 | 0.0 | 0.0 | 0.0 | 0.0 | 0.0 |
| | Gaussian [11] | 1.553 | 99.2 | 98.3 | 96.8 | 94.3 | 89.7 | 81.9 | 67.3 | 43.6 | 0.0 | 0.0 | 0.0 | 0.0 |
| | Stability training [38] | 1.570 | 99.2 | 98.5 | 97.1 | 94.8 | 90.7 | 83.2 | 69.2 | 45.4 | 0.0 | 0.0 | 0.0 | 0.0 |
| | SmoothAdv [49] | 1.687 | 99.0 | 98.3 | 97.3 | 95.8 | 93.2 | 88.5 | 81.1 | 67.5 | 0.0 | 0.0 | 0.0 | 0.0 |
| | MACER [65] | 1.594 | 98.5 | 97.5 | 96.2 | 93.7 | 90.0 | 83.7 | 72.2 | 54.0 | 0.0 | 0.0 | 0.0 | 0.0 |
| 0.50 | Consistency [23] | 1.657 | 99.2 | 98.6 | 97.6 | 95.9 | 93.0 | 87.8 | 78.5 | 60.5 | 0.0 | 0.0 | 0.0 | 0.0 |
| | **SmoothMix** ($\eta = 1.0$) | **1.678** | 99.0 | **98.4** | **97.4** | **95.7** | **93.0** | **88.1** | **80.0** | **65.6** | 0.0 | 0.0 | 0.0 | 0.0 |
| | **+ One-step adversary** | **1.694** | 98.8 | 98.1 | **97.1** | 95.3 | 92.7 | **88.3** | **81.7** | **69.5** | 0.0 | 0.0 | 0.0 | 0.0 |
| | **SmoothMix** ($\eta = 5.0$) | **1.694** | 98.7 | 98.0 | **97.0** | 95.3 | 92.7 | **88.5** | **81.8** | **70.0** | 0.0 | 0.0 | 0.0 | 0.0 |
| | **+ One-step adversary** | **1.685** | 98.2 | 97.5 | 96.3 | **94.5** | 91.3 | 87.4 | 81.0 | **70.7** | 0.0 | 0.0 | 0.0 | 0.0 |
| | Gaussian [11] | 1.620 | 96.3 | 94.4 | 91.4 | 86.8 | 79.8 | 70.9 | 59.4 | 46.2 | 32.5 | 19.7 | 10.9 | 5.8 |
| | Stability training [38] | 1.634 | 96.5 | 94.6 | 91.6 | 87.2 | 80.7 | 71.7 | 60.5 | 47.0 | 33.4 | 20.6 | 11.2 | 5.9 |
| | SmoothAdv [49] | 1.779 | 95.8 | 93.9 | 90.6 | 86.5 | 80.8 | 73.7 | 64.6 | 53.9 | 43.3 | 32.8 | 22.2 | 12.1 |
| | MACER [65] | 1.598 | 91.6 | 88.1 | 83.5 | 77.7 | 71.1 | 63.7 | 55.7 | 46.8 | 38.4 | 29.2 | 20.0 | 11.5 |
| 1.00 | Consistency [23] | 1.740 | 95.0 | 93.0 | 89.7 | 85.4 | 79.7 | 72.7 | 63.6 | 53.0 | 41.7 | 30.8 | 20.3 | 10.7 |
| | **SmoothMix** ($\eta = 1.0$) | **1.788** | 95.5 | 93.5 | 90.5 | 86.2 | **80.6** | **73.4** | **64.3** | **53.7** | **43.2** | **33.5** | **23.9** | **14.1** |
| | **+ One-step adversary** | **1.816** | 94.7 | 92.4 | 89.2 | 84.6 | 79.4 | **72.5** | **64.0** | **54.5** | **44.8** | **36.2** | **27.4** | **18.7** |
| | **SmoothMix** ($\eta = 5.0$) | **1.820** | 93.7 | 91.6 | 88.1 | 83.5 | 77.9 | 70.9 | **62.7** | 53.8 | **44.8** | **36.6** | **28.9** | **21.5** |
| | **+ One-step adversary** | **1.823** | 93.3 | 90.9 | 87.5 | 83.0 | 77.5 | 70.6 | **62.7** | 53.4 | **44.9** | **37.1** | **29.3** | **22.4** |

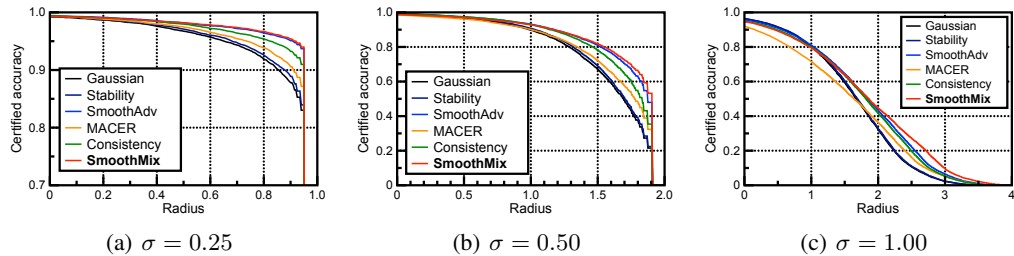

(a) $\sigma = 0.25$  (b) $\sigma = 0.50$  (c) $\sigma = 1.00$

Figure 4: Comparison of approximate certified accuracy for various training methods on MNIST. The sharp drop of certified accuracy in each plot is due to that there is a strict upper bound in radius that CERTIFY can output for a given $\sigma$ and $n = 100,000$.

assigns 0 for the incorrectly classified test samples, *i.e.*, when $\hat{f}(x) \neq y$, so that a decreased clean accuracy of $\hat{f}$ would negatively affect the value of ACR. We use the official PyTorch implementation[8] of CERTIFY, with $n = 100,000$, $n_0 = 100$ and $\alpha = 0.001$, following [11, 49, 23].

## 4.1 Results on MNIST

For MNIST [32] experiments, we report the approximate certified accuracy and ACR of smoothed classifiers obtained from LeNet [32] with different training methods, including SmoothMix, using the full MNIST test dataset. We consider three different models as varying the noise level $\sigma \in \{0.25, 0.5, 1.0\}$. During inference, we apply randomized smoothing with the same $\sigma$ used in the training. When SmoothMix is used, we consider a fixed hyperparameter value for $\alpha = 1.0$ and

[8]https://github.com/locuslab/smoothing

Table 3: Comparison of approximate certified test accuracy (%) and ACR on CIFAR-10. We set our results bold-faced whenever the value improves the Gaussian baseline, and underlined whenever the value improves the best among the considered baselines. $*$ indicates that the results are evaluated from the official pre-trained models released by authors.

| $\sigma$ | Models (CIFAR-10) | ACR | 0.00 | 0.25 | 0.50 | 0.75 | 1.00 | 1.25 | 1.50 | 1.75 |
|---|---|---|---|---|---|---|---|---|---|---|
| | Gaussian [11] | 0.424 | 76.6 | 61.2 | 42.2 | 25.1 | 0.0 | 0.0 | 0.0 | 0.0 |
| | Stability training [38] | 0.421 | 72.3 | 58.0 | 43.3 | 27.3 | 0.0 | 0.0 | 0.0 | 0.0 |
| | SmoothAdv$*$ [49] | 0.544 | 73.4 | 65.6 | 57.0 | 47.5 | 0.0 | 0.0 | 0.0 | 0.0 |
| 0.25 | MACER$*$ [65] | 0.531 | 79.5 | 69.0 | 55.8 | 40.6 | 0.0 | 0.0 | 0.0 | 0.0 |
| | Consistency [23] | 0.552 | 75.8 | 67.6 | 58.1 | 46.7 | 0.0 | 0.0 | 0.0 | 0.0 |
| | **SmoothMix (Ours)** | 0.553 | **77.1** | **67.9** | **57.9** | **46.7** | 0.0 | 0.0 | 0.0 | 0.0 |
| | **+ One-step adversary** | 0.548 | 74.2 | 66.1 | 57.4 | 47.7 | 0.0 | 0.0 | 0.0 | 0.0 |
| | Gaussian [11] | 0.525 | 65.7 | 54.9 | 42.8 | 32.5 | 22.0 | 14.1 | 8.3 | 3.9 |
| | Stability training [38] | 0.521 | 60.6 | 51.5 | 41.4 | 32.5 | 23.9 | 15.3 | 9.6 | 5.0 |
| | SmoothAdv$*$ [49] | 0.684 | 65.3 | 57.8 | 49.9 | 41.7 | 33.7 | 26.0 | 19.5 | 12.9 |
| 0.50 | MACER$*$ [65] | 0.691 | 64.2 | 57.5 | 49.9 | 42.3 | 34.8 | 27.6 | 20.2 | 12.6 |
| | Consistency [23] | 0.720 | 64.3 | 57.5 | 50.6 | 43.2 | 36.2 | 29.5 | 22.8 | 16.1 |
| | **SmoothMix (Ours)** | **0.715** | 65.0 | **56.7** | **49.2** | **41.2** | **34.5** | 29.6 | 23.5 | 18.1 |
| | **+ One-step adversary** | 0.737 | 61.8 | **55.9** | **49.5** | 43.3 | 37.2 | 31.7 | 25.7 | 19.8 |

$m = 4$, the step size and the number of noise samples. We empirically observe that it is beneficial to set $\alpha \cdot T$ to be proportional to $\sigma$, the noise level, as there exist different upper bounds on the certified radius statistical achievable in practice depending on $\sigma$: in this respect, we set $T = 2, 4, 8$ for the models with $\sigma = 0.25, 0.5, 1.0$, respectively. We apply the same $m = 4$ for SmoothAdv, i.e., for adversarial training, as well, and $T = 10$ with an $\ell_2$-constraint of radius $\varepsilon = 1.0$. Here, notice that SmoothMix and SmoothAdv use the same number of hyperparameters: more specifically, although SmoothMix introduces $\alpha$ compared to SmoothAdv, the step size in (8), it instead does not use the hyperparameter $\varepsilon$ of SmoothAdv, i.e., the maximum norm of adversarial perturbations.

The results are presented in Table 2 and Figure 4. Overall, we observe that our proposed SmoothMix loss (10) added to the Gaussian training dramatically improves the certified test accuracy from "Gaussian". By considering the one-step adversary (Section 3.2) in training, we could further improve the robust accuracy, significantly improving ACRs compared to the previous state-of-the-art training methods: e.g., our method could improve ACRs with $\sigma = 1.0$ from $1.779 \rightarrow 1.823$. This shows that improvements from SmoothMix can be orthogonal to those from SmoothAdv. It is also remarkable that even without the one-step adversarial example, one could further improve the certified robustness by simply increasing the relative strength $\eta$ of the SmoothMix loss, e.g., by $1.0 \rightarrow 5.0$ as presented in Table 2: e.g., "SmoothMix" with $\eta = 5.0$ still outperforms "SmoothAdv" by $1.779 \rightarrow 1.820$ at $\sigma = 1.0$. Finally, we note that our models could substantially improve the robustness at larger perturbations with less degradation in the clean accuracy, e.g., compared to "MACER" or "Consistency": considering that they are also regularization based approaches that allow to control the robustness via controlling their regularization strength, the results show that our form of loss could better compensate the trade-off between accuracy and robustness.

## 4.2 Results on CIFAR-10

For CIFAR-10 [29] experiments, we report the approximate certified accuracy and ACR of smoothed classifiers from ResNet-110 [20] using the full CIFAR-10 test dataset. Again, we consider three different models as varying the noise level $\sigma \in \{0.25, 0.5, 1.0\}$,[9] and apply the same $\sigma$ for inference as well. When SmoothMix is used, we consider a fixed hyperparameter value for $T = 4$ and $m = 2$, the number of steps and the number of noise samples, respectively. We also fix $\eta = 5.0$ throughout the experiments, as also used for MNIST (see Table 2). Again, we make sure that $\alpha \cdot T$ to be proportional to $\sigma$, so that we set $\alpha = 0.5, 1.0, 2.0$ for the models with $\sigma = 0.25, 0.5, 1.0$, respectively. For SmoothAdv, we report the performance evaluated from the pre-trained models released by the authors[10] for a fixed configuration of $T = 10, \varepsilon = 1.0$, and $m = 8$.

---

[9]Due to the space limitation, we defer the CIFAR-10 results with $\sigma = 1.0$ to Appendix F, considering that the scenario can be less practical compared to the others: e.g., the clean accuracy in this setup is $< 50\%$ in most cases, even for the Gaussian baseline [11].

[10]https://github.com/Hadisalman/smoothing-adversarial

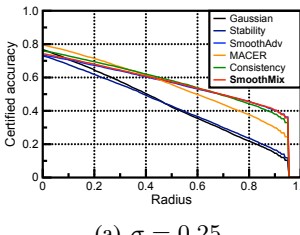
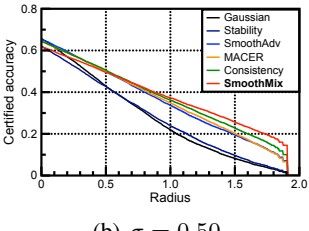
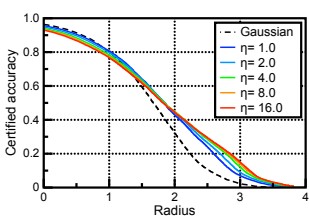

(a) $\sigma = 0.25$       (b) $\sigma = 0.50$

Figure 5: Comparison of approximate certified accuracy for various training methods on CIFAR-10. The sharp drop of certified accuracy in each plot is due to that there is a strict upper bound in radius that CERTIFY can output for a given $\sigma$ and $n = 100,000$.

Figure 6: Comparison of approximate certified test accuracy of SmoothMix for varying $\eta$. "Gaussian" indicates the baseline training with Gaussian augmentation.

The results are summarized in Table 3 and Figure 5. Again, we still observe that our method generally exhibits better trade-offs between accuracy and certified robustness compared to other baselines: *e.g.*, at $\sigma = 0.5$, "SmoothMix" could improve the previous best result from "Consistency" by a significant margin of $0.720 \rightarrow 0.737$. Without the single-step adversary, "SmoothMix" can effectively preserve the clean accuracy while also improving ACR, *e.g.*, at $\sigma = 0.25$, "SmoothMix" could even improve the clean accuracy of "Gaussian": although "MACER" could improve the clean accuray as well, one could see that their improvements in robust accuracy are relatively limited. It is also notable that the *certified test accuracy* we report in Table 3 can sometimes complement ACR: although "Consistecny" achieves a competitive ACR with "SmoothMix" at $\sigma = 0.25$, one can still confirm the superiority of "SmoothMix" by comparing the certified accuracy $r = 0.0$ (*i.e.*, the clean accuracy), namely 75.8% vs. 77.1%, given that they both achieve similar certified accuracy at $r = 0.75$ (*i.e.*, the robust accuracy). This is because a bare increase in the clean accuracy (*i.e.*, $\hat{f}$ correctly classifies more test samples but with CR's closer to 0) often contributes less to the increase in ACR.

## 4.3 Ablation study

We also conduct an ablation study to investigate the individual effects of the hyperparameters in our method. Unless otherwise noted, we perform experiments on MNIST with $\sigma = 1.0$. All the detailed results from this ablation study are reported in Appendix H.

**Equal-confidence mixing ratios.** Recall from Figure 2 that we are motivated by the problem of *miscalibration* in smoothed classifiers between clean and its adversarial example. To see how much the proposed SmoothMix could alleviate this issue, we compare the distributions of the minimal mixing ratios that changes its prediction of a given classifier on the CIFAR-10 test samples, namely the *equal-confidence mixing ratios*, before and after training with SmoothMix. We find the adversarial examples separately from two pre-trained ResNet-110 based (smoothed) classifiers, each trained by Gaussian training and SmoothMix, respectively, assuming $\sigma = 0.25$. We optimize each adversarial example assuming only a quite loose norm-bound of $\varepsilon = 8$ to allow more update steps, *i.e.*, via 50-step PGD for both classifiers. Figure 3 shows the result, and it indeed confirms SmoothMix has an effect of improving calibration between clean and adversarial examples.

**Effect of $\eta$.** By design, SmoothMix controls the trade-off between accuracy and robustness by adjusting $\eta$, the relative strength of $L^{\texttt{mix}}$ over $L^{\texttt{nat}}$ (11). Here, we further examine the effect of $\eta$ by comparing the certified robustness on varying $\eta \in \{1, 2, 4, 8, 16\}$: the results in Figure 6 show that increasing $\eta$ consistently improves the certified robustness of the classifier, which confirms $L^{\texttt{mix}}$, the mixup loss, as an effective term to trade-off the robustness against $L^{\texttt{nat}}$ for accuracy.

**Trade-off between $\alpha$ and $T$.** In practice, SmoothMix can trade-off between the step size $\alpha$ and the number of steps $T$ to compensate between a more accurate optimization of (7) and its computational cost, while maintaining the effective range of the perturbation by $\alpha \cdot T$. Figure 7(a) explores this trade-off, by comparing models trained with different combinations of $(\alpha, T)$ under control of $\alpha \cdot T = 8.0$. Interestingly, the results indicate that the choice of $\alpha$ and $T$ does not significantly affect the final performance as long as $\alpha \cdot T$ is constant: all the considered combinations achieve similar robustness, with only a slight degradation in ACR even at $(\alpha, T) = (8.0, 1)$ (see Table 9 in Appendix H). This

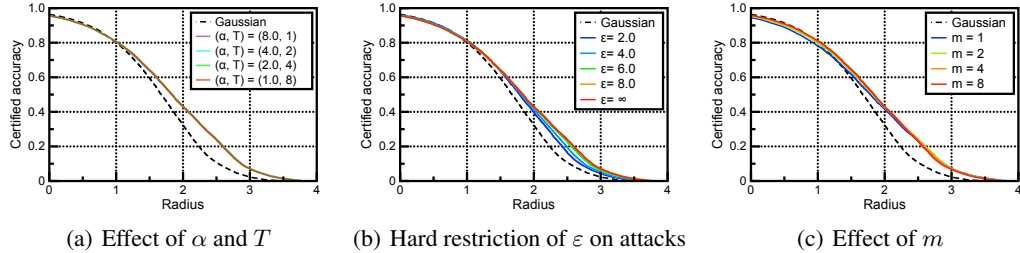

|  (a) Effect of $\alpha$ and $T$ | (b) Hard restriction of $\varepsilon$ on attacks | (c) Effect of $m$ |

Figure 7: Comparison of approximate certified test accuracy of SmoothMix and its ablations. "Gaussian" indicates the baseline training with Gaussian augmentation.

suggests that (a) finding adversarial examples in a smoothed classifier can be simpler than one might expect, and (b) one can effectively reduce the training cost of SmoothMix using small $T$ in practice.

**Hard restriction on adversarial attacks.** One of key features of SmoothMix is at its *unrestricted* search of adversarial examples. Here, we examine the case when there is a hard restriction on each search, namely in $\ell_2$-radius of $\varepsilon \in \{2, 4, 6, 8\}$. The results presented in Figure 7(b) along with the Gaussian baseline ("Gaussian") and the original unrestricted setup ("$\varepsilon = \infty$") show that SmoothMix indeed works best when there is no such restrictions, although these ablations still reasonably improve the Gaussian baseline, *i.e.*, calibrating with adversarial examples outside the $\varepsilon$-ball can indeed help to improve the certified robustness in our training scheme.

**Effect of $m$.** Figure 7(c) (and Table 11 in Appendix H) investigates the effect of using different $m \in \{1, 2, 4, 8\}$, the number of noise samples to approximate the prediction of smoothed classifier: the larger $m$, the better approximation of smoothed classifier, which would be beneficial for both natural loss and SmoothMix loss (11). Overall, we observe that SmoothMix can still improve ACR from "Gaussian" even with $m = 1$, but with a moderate degradation in the clean accuracy: as $m$ is one of the crucial factors related to the total training cost in practice, one is recommended to use smaller $m$, *e.g.*, $m = 2$ or $4$, considering its little effect to the final ACR.

## 5 Discussion and conclusion

We observe that adversarial training with an *unrestricted* adversary can be feasible and even more promising (compared to the *restricted* ones) when it comes with smoothed classifiers, by showing their effectiveness to improve the certified adversarial robustness with a novel mixup-based training. We address the brittleness of deep neural networks through the lens of smoothed classifiers, which could give us a simpler view on them. We believe our research could be a useful step toward understanding what essentially constitutes adversarial examples in deep neural networks.

**Broader impact.** Adversarial robustness in deep learning is arguably an essential requirement for *AI safety* [2], with much impact on various security-concerned systems: *e.g.*, medical diagnosis [9], speech recognition [46] and autonomous driving [64]. Thanks to their certifiable guarantees, we believe a practical success of systems based on *randomized smoothing* would be fatal for those who maliciously attempt to break down the system via adversarial attacks. Nevertheless, one should also recognize that current techniques for robustness in deep learning, including randomized smoothing as well, indeed have a clear gap to be practically used in real-world, *e.g.*, defending against challenging unrestricted attacks [60, 5], which should be further investigated in the future research. Consequently, it is particularly important for the defense techniques not to be misused in practical systems, as a failure of such systems may lead practitioners to have a biased, false sense of security.

**Limitation.** While our ultimate goal is to find optimal smoothed classifiers in terms of the accuracy and robustness trade-off, SmoothMix should not be considered as the final solution for the problem; our method is a promising proof-of-concept showing the close relationship between randomized smoothing and *confidence-calibrated* classifiers [19, 36]. Although our focus in this paper is currently limited only to the over-confidence issue, we believe there are still many rooms to be explored in future for another such connection, *e.g.*, could the recent developments in the literature of uncertainty estimation of deep neural networks [22, 54] help to improve the robustness of smoothed classifiers.

## Acknowledgments and Disclosure of Funding

This work was conducted by Center for Applied Research in Artificial Intelligence (CARAI) grant funded by Defense Acquisition Program Administration (DAPA) and Agency for Defense Development (ADD) (UD190031RD). The authors would like to thank Jaeho Lee for helpful discussions.

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
