## A Training procedure of SmoothMix

---

**Algorithm 1** SmoothMix training

---

**Input:** Sample $(x, y) \sim P$. smoothing factor $\sigma$. number of noise samples $m$. number of steps $T$. step size $\alpha$. regularization strength $\eta > 0$.

---

1: Sample $\delta_1, \cdots, \delta_m \sim \mathcal{N}(0, \sigma^2 I)$, and $\lambda \sim \mathcal{U}([0, \frac{1}{2}])$
2: // FIND AN ADVERSARIAL EXAMPLE
3: $\tilde{x}^{(0)}, \hat{F}(x^{(0)}) \leftarrow x, \frac{1}{m} \sum_{i=1}^{m} F(x + \delta_i)$
4: **for** $t = 0$ **to** $T - 1$ **do**
5:      $J(\tilde{x}^{(t)}) \leftarrow -\log \hat{F}_y(\tilde{x}^{(t)})$
6:      $\tilde{x}^{(t+1)} \leftarrow \tilde{x}^{(t)} + \alpha \cdot \frac{\nabla_x J(\tilde{x}^{(t)})}{\|\nabla_x J(\tilde{x}^{(t)})\|_2}$
7:      $\hat{F}(\tilde{x}^{(t+1)}) \leftarrow \frac{1}{m} \sum_{i=1}^{m} F(\tilde{x}^{(t+1)} + \delta_i)$
8: **end for**
9: **if** use_single_step **then** $x \leftarrow \tilde{x}^{(1)}$
10: // COMPUTE THE SMOOTHMIX LOSS
11: $x^{\texttt{mix}}, y^{\texttt{mix}} \leftarrow ((1 - \lambda) \cdot x + \lambda \cdot \tilde{x}^{(T)}), ((1 - \lambda) \cdot \hat{F}(x) + \lambda \cdot \frac{1}{C})$
12: **for** $i = 1$ **to** $m$ **do**
13:      $L_i^{\texttt{nat}}, L_i^{\texttt{mix}} \leftarrow \mathcal{L}(F(x + \delta_i), y), \mathcal{L}(F(x^{\texttt{mix}} + \delta_i), y^{\texttt{mix}})$
14: **end for**
15: $L \leftarrow \frac{1}{m} \sum_i (L_i^{\texttt{nat}} + \eta \cdot L_i^{\texttt{mix}})$

---

## B Discussion on input-dependent designs of noise scales

In this paper, we aim to develop a new training method to exhibit a better trade-off between accuracy and (certified) robustness of smoothed classifiers. Meanwhile, there has been recently another proposal to improve the robustness of smoothed classifiers without a new training scheme, namely by certifying a given smoothed classifier with *input-dependent* $\sigma$ [1, 58, 10]. In this section, however, we show that if one allows *different* noise scales $\sigma$ for each input in attempt to generalize the current framework of randomized smoothing [11], then the actual robustness guarantee would rapidly decrease as the input dimension grows. In particular, we consider the following classifier $\tilde{f}$ generalizing (2) with some non-negative function $g : \mathbb{R}^d \to \mathbb{R}_{\geq 0}$, defined as follows:

$$\tilde{f}(x) := \arg\max_{c \in \mathcal{Y}} \mathbb{P}_{\delta \sim \mathcal{N}(0, g(x)I)}(f(x + \delta) = c),$$

In other words, we assume that the scaling parameter of the smoothing noise can now be a function of $x$. As in the main text, we are interested in the certified radius $\underline{R}(\hat{f}; x, y)$ of $\tilde{f}$.

One may expect that $\underline{R}(\tilde{f}; x, y)$ can be significantly larger than $\underline{R}(\hat{f}; x, y)$ since $\hat{f}$ is a special case of $\tilde{f}$, *i.e.*, constant $g(x)$. However, we show that it may not be true for high-dimensional inputs: even a small deviation of $g(x)$ can incur very poor certified robustness. Formally, we prove the following theorem.

**Theorem 1.** *Let $r_i, i \in \mathbb{N}$ be any i.i.d. random variables of zero mean, unit variance, and $\mathbb{E}[r_i^4] < \infty$. Let $\mathcal{F}_d$ be a collection of all measurable functions from $\mathbb{R}^d$ to $\{0, 1\}$. Let $p \in (0.5, 1)$, $\sigma, \tau > 0$, and $\varepsilon \in (0, 1/2]$ be constants such that $\sigma \neq \tau$. Then, for $\delta := (r_1, \ldots, r_d)$, for any $c \in \{0, 1\}$, and for any $d \in \mathbb{N}$, the following statements hold:*

$$\sup_{x, x' \in \mathbb{R}^d : \|x - x'\|_2 \leq \varepsilon} \quad \inf_{f \in \mathcal{F}_d : \mathbb{P}(f(x + \sigma\delta) = c) = p} \mathbb{P}(f(x' + \tau\delta) = c) \leq C/d.$$

*for some constant $C > 0$ which is a function of other constants $p, \sigma, \tau, \varepsilon, \mathbb{E}[r_i^4]$.*

Theorem 1 indicates the curse of dimensionality for the worst classifier under general noises of a finite kurtosis. In particular, it states that there exists an upper bound on $\mathbb{P}(f(x + \tau\delta) = c)$ inversely proportional to the input dimension $d$ even though two inputs $x, x'$ are extremely close. Hence, if we utilize different noise scales (*i.e.*, $\sigma$) for each input, the resulting lower bound on the certified radius relying on the worst-case bound as in [11, 33, 49] will be small for high-dimensional inputs. Namely, choosing (almost) constant noise scale for the inputs in the target certification region is necessary.

## B.1 Proof of Theorem 1

We first define $z = (z_1, \ldots, z_d) := x' - x$, *i.e.*, $\|z\|_2 \leq \varepsilon$. Then, the following inequality trivially holds.

$$\inf_{f \in \mathcal{F}_d : \mathbb{E}[f(x+\sigma\delta)]=p} \mathbb{E}[f(x'+\tau\delta)] \leq \inf_{\mathcal{U} \subset \mathbb{R}^d : \mathbb{P}(\sigma\delta \in \mathcal{U})=p} \mathbb{P}(\tau\delta + z \in \mathcal{U})$$

$$\leq \mathbb{P}\left(\frac{\|\tau\delta + z\|_2^2}{d} \in [\sigma^2 - k, \sigma^2 + k]\right) \quad (12)$$

where $k$ is a non-negative number satisfying

$$\mathbb{P}\left(\frac{\|\sigma\delta\|_2^2}{d} \in [\sigma^2 - k, \sigma^2 + k]\right) = p.$$

The following lemma asserts that the RHS of (12) is bounded by $C/d$ where $C$ is some constant which is only a function of $\mathbb{E}[r_i^4], \sigma, \tau, \varepsilon, p$. This completes the proof of Theorem 1.

**Lemma 2.** *There exists $C$ which is a function of $\mathbb{E}[r_i^4], \sigma, \tau, \varepsilon, p$ such that the following statements hold: for any $d \in \mathbb{N}$ and for any $z \in \mathbb{R}^d$ satisfying $\|z\|_2 \leq \varepsilon$,*

$$\mathbb{P}\left(\frac{\|\tau\delta + z\|_2^2}{d} \in [\sigma^2 - k, \sigma^2 + k]\right) \leq \frac{C}{d}.$$

## B.2 Proof of Lemma 2

Lemma 2 is a direct consequence of the law of large numbers applied to the i.i.d. random variables $r_i^2$. First, we compute the variance of $\frac{\|\sigma\delta\|_2^2}{d}$ using the following equality: for $\eta := \sqrt{\mathbb{E}[r_i^4] - 1}$,

$$\text{Var}\left(\frac{\|\sigma\delta\|_2^2}{d}\right) = \mathbb{E}\left[\left(\frac{\|\sigma\delta\|_2^2}{d} - \sigma^2\right)^2\right] = \mathbb{E}\left[\left(\frac{\sigma^2}{d}\sum_{i=1}^{d}(r_i^2 - 1)\right)^2\right]$$

$$= \frac{\sigma^4}{d^2}\sum_{i=1}^{d}\mathbb{E}\left[(r_i^2 - 1)^2\right] = \frac{\sigma^4}{d^2}\sum_{i=1}^{d}\mathbb{E}[r_i^4] - 1$$

$$= \frac{\sigma^4(\mathbb{E}[r_i^4] - 1)}{d} = \frac{\sigma^4\eta^2}{d}$$

where the third equality follows from the independence of $r_i$s and the fourth inequality follows from $\mathbb{E}[r_i^2] = 1$. Hence, from the Chebyshev's inequality, we have

$$\mathbb{P}\left(\left|\frac{\|\sigma\delta\|_2^2}{d} - \sigma^2\right| < \frac{\sigma^2\eta}{\sqrt{d(1-p)}}\right) \geq 1 - (\sqrt{1-p})^2 = p, \quad (13)$$

*i.e.*, $k \leq \frac{\sigma^2\eta}{\sqrt{d(1-p)}}$.

Now, we derive a similar concentration inequality for $\frac{\|\tau\delta + z\|_2^2}{d}$. To this end, we bound its deviation from $\tau^2 + \frac{\|z\|_2^2}{d}$ as follows:

$$\mathbb{P}\left(\left|\frac{\|\tau\delta + z\|_2^2}{d} - \left(\tau^2 + \frac{\|z\|_2^2}{d}\right)\right| \geq \frac{\sigma^2 + \tau^2}{3}\right)$$

$$= 1 - \mathbb{P}\left(\left|\frac{\|\tau\delta + z\|_2^2}{d} - \left(\tau^2 + \frac{\|z\|_2^2}{d}\right)\right| < \frac{\sigma^2 + \tau^2}{3}\right)$$

$$\leq 1 - \mathbb{P}\left(\left|\frac{\|\tau\delta\|_2^2}{d} - \tau^2\right| < \frac{\sigma^2 + \tau^2}{6} \quad \text{and} \quad \left|\frac{2\tau\sum_{i=1}^d r_i z_i}{d}\right| < \frac{\sigma^2 + \tau^2}{6}\right)$$

$$= \mathbb{P}\left(\left|\frac{\|\tau\delta\|_2^2}{d} - \tau^2\right| \geq \frac{\sigma^2 + \tau^2}{6} \quad \text{or} \quad \left|\frac{2\tau\sum_{i=1}^d r_i z_i}{d}\right| \geq \frac{\sigma^2 + \tau^2}{6}\right)$$

$$\leq \mathbb{P}\left(\left|\frac{\|\tau\delta\|_2^2}{d} - \tau^2\right| \geq \frac{\sigma^2 + \tau^2}{6}\right) + \mathbb{P}\left(\left|\frac{2\tau\sum_{i=1}^d r_i z_i}{d}\right| \geq \frac{\sigma^2 + \tau^2}{6}\right)$$

$$\leq \frac{36\tau^4\eta^2 + 144\tau^2\varepsilon^2}{(\sigma^2 + \tau^2)^2 d} \tag{14}$$

where the last inequality is from the variance bounds

$$\mathrm{Var}\left(\frac{\|\tau\delta\|_2^2}{d} - \tau^2\right) = \frac{\tau^4\eta^2}{d}$$

$$\mathrm{Var}\left(\frac{2\tau\sum_{i=1}^d r_i z_i}{d}\right) = \frac{4\tau^2\|z\|_2^2}{d^2} \leq \frac{4\tau^2\varepsilon^2}{d^2}$$

and the Chebyshev's inequality

$$\mathbb{P}\left(\left|\frac{\|\tau\delta\|_2^2}{d} - \tau^2\right| \geq \frac{\sigma^2 + \tau^2}{6}\right) \leq \frac{36\tau^4\eta^2}{(\sigma^2 + \tau^2)^2 d}$$

$$\mathbb{P}\left(\left|\frac{2\tau\sum_{i=1}^d r_i z_i}{d}\right| \geq \frac{\sigma^2 + \tau^2}{6}\right) \leq \frac{144\tau^2\varepsilon^2}{(\sigma^2 + \tau^2)^2 d^2}.$$

Then, for all $d \geq \max\left\{\frac{4\sigma^4\eta^2}{(\tau^2 - \sigma^2)^2(1-p)}, \frac{6\varepsilon^2}{\sigma^2 + \tau^2}\right\}$, *i.e.*, $\frac{\sigma^2\eta}{\sqrt{d(1-p)}} \leq \frac{|\tau^2 - \sigma^2|}{2}$ and $\frac{\varepsilon^2}{d} \leq \frac{\sigma^2 + \tau^2}{6}$, it holds that

$$\mathbb{P}\left(\frac{\|\tau\delta + z\|_2^2}{d} \in [\sigma^2 - k, \sigma^2 + k]\right)$$

$$\leq \mathbb{P}\left(\frac{\|\tau\delta + z\|_2^2}{d} \in \left[\sigma^2 - \frac{|\tau^2 - \sigma^2|}{2}, \sigma^2 + \frac{|\tau^2 - \sigma^2|}{2}\right]\right)$$

$$\leq \mathbb{P}\left(\left|\frac{\|\tau\delta + z\|_2^2}{d} - \left(\tau^2 + \frac{\|z\|_2^2}{d}\right)\right| \geq \frac{\sigma^2 + \tau^2}{3}\right)$$

$$\leq \frac{36\tau^4\eta^2 + 144\tau^2\varepsilon^2}{(\sigma^2 + \tau^2)^2 d}$$

by using (14). Hence, choosing

$$C := \max\left\{\frac{4\sigma^4\eta^2}{(\tau^2 - \sigma^2)^2(1-p)}, \frac{6\varepsilon^2}{\sigma^2 + \tau^2}, \frac{36\tau^4\eta^2 + 144\tau^2\varepsilon^2}{(\sigma^2 + \tau^2)^2}\right\}$$

completes the proof of Lemma 2.

# C Experimental details

Throughout our experiments, we follow the same training details of prior works [11, 49, 65, 23] for a fair comparison: more specifically, we use LeNet [32] for MNIST, ResNet-110 [20] for CIFAR-10, and ResNet-50 [20] for ImageNet. We train every model via stochastic gradient descent using Nesterov momentum of weight 0.9 without dampening. We set a weight decay of $10^{-4}$ for all the models. We consider three different noise levels $\sigma \in \{0.25, 0.5, 1.0\}$ for smoothing classifiers for MNIST and CIFAR-10 models, and $\sigma \in \{0.5, 1.0\}$ in the case of ImageNet. We used up to 4 NVIDIA TITAN Xp GPUs to run each configurations considered in our experiments, both for training and certification: more specifically, we used a single GPU to run every experimenet on MNIST and CIFAR-10, and four GPUs to run ImageNet models.

## C.1 Datasets

**MNIST** dataset [32] consists 70,000 gray-scale hand-written digit images of size 28×28, 60,000 for training and 10,000 for testing. Each of the images is labeled from 0 to 9, i.e., there are 10 classes. We do not perform any pre-processing except for normalizing the range of each pixel from 0-255 to 0-1. When MNIST is used for training, we use LeNet [32] for 90 epochs and use the initial learning rate of 0.01. The learning rate is decayed by 0.1 at 30-th and 60-th epoch.

**CIFAR-10** dataset [29] consist of 60,000 RGB images of size 32×32 pixels, 50,000 for training and 10,000 for testing. Each of the images is labeled to one of 10 classes, and the number of data per class is set evenly, i.e., 6,000 images per each class. We use the standard data-augmentation scheme of random horizontal flip and random translation up to 4 pixels, as also used by other baselines [11, 49, 65, 23]. We also normalize the images in pixel-wise by the mean and the standard deviation calculated from the training set. When CIFAR-10 is used for training, we train ResNet-110 [20] models for 150 epochs with initial learning rate of 0.1. The learning rate if decated by 0.1 at 50-th and 100-th epoch.

**ImageNet** classification dataset [48] consists of 1.2 million training images and 50,000 validation images, which are labeled by one of 1,000 classes. For data-augmentation, we perform 224×224 random cropping with random resizing and horizontal flipping to the training images. At test time, on the other hand, 224×224 center cropping is performed after re-scaling the images into 256×256. When ImageNet is used for training, we train ResNet-50 [20] models for 90 epochs with initial learning rate of 0.1. The learning rate if decated by 0.1 at 30-th and 60-th epoch.

## C.2 Detailed hyperparameters for baselines

**Stability training [38]** uses a single hyperparameter $\lambda > 0$ to control the relative strength of the stability regularization compared to the standard cross-entropy loss. In our experiments, we use $\lambda = 2$ by default for this method, but except for the "$\sigma = 1.0$" model on CIFAR-10: in this case, we had to reduce it to $\lambda = 1$ for a stable training.

**SmoothAdv [49]** mainly controls three hyperparameters those are for performing projected gradient descent (PGD) to find adversarial examples in the training: namely, it uses $m$: the number of noise samples, $T$: the number of PGD steps, and $\varepsilon$: an $\ell_2$-norm restriction on adversarial perturbations. For SmoothAdv models, we fix $T = 10$ and $\varepsilon = 1.0$ throughout the experiments. In case of $m$, and use $m = 4$ for MNIST models, and $m = 8$ for CIFAR-10. Following Salman et al. [49], we also adopt the *warm-up* strategy on $\varepsilon$, *i.e.*, it is initially set to zero, and gradually increased for the first 10 epochs up to the original value of $\varepsilon$.

**MACER [65]** adds four hyperparameters to the training: namely, it uses $m$: the number of noise samples, $\lambda$: the relative strength of regularization, $\beta$: a temperature scaling factor, and $\gamma$: a margin gap. We follow the configurations reported by Zhai et al. [65] to reproduce the MNIST results: namely, we use $m = 16$, $\beta = 16.0$, $\gamma = 8.0$ and $\lambda = 16.0$. We use $\lambda = 6.0$ in case of $\sigma = 1.0$ on MNIST, however, for a better training stability. We use the pre-trained models released by the authors for evaluations on CIFAR-10, which can be downloaded at `https://github.com/RuntianZ/macer`. These CIFAR-10 models are reported to be trained with $m = 16$, $\beta = 16.0$, $\gamma = 8.0$, and $\lambda = 12.0$ and 4.0 for $\sigma = 0.25$ and 0.5, respectively. For $\sigma = 1.0$, $\lambda$ is initially set to 0, and changed to $\lambda = 12.0$ after the first learning rate decay.

**Consistency [23]** controls two hyperparameters, namely $\lambda$ and $\eta$, each for the relative strength of the consistency term and the entropy term, respectively. We obtain results from the best hyperparameters those reported by Jeong and Shin [23] when the consistency regularization is applied to the Gaussian training baseline, both in MNIST and CIFAR-10 datasets. More concretely, we fix $\eta = 0.5$ for every model, and use $\lambda = 5$ for MNIST and $\lambda = 10$ for CIFAR-10 models by default. In case of $\sigma = 0.25$, $\lambda$ is doubled in both datasets, *i.e.*, $\lambda = 10$ and $\lambda = 20$ for MNIST and CIFAR-10, respectively, as it is shown to achieve better ACRs.

## D    Related work

**Certified adversarial robustness.**   We focus on improving adversarial robustness of *randomized smoothing* [11] based classifiers, which is currently one of prominent ways to obtain a classifier with a robustness certification. In general, there have been many attempts other than randomized smoothing to provide a robustness certification of deep neural networks [16, 59, 42, 61, 18, 68], and correspondingly with attempts to further improve the robustness with respect to those certification protocols [13, 12, 4]. Nevertheless, randomized smoothing has attracted particular attention as the first approach that could successfully scaled up to the ImageNet dataset [48]. A more complete taxonomy on the literature can be found in Li et al. [39].

**Confidence-calibrated training.**   *Overconfident predictions* of deep neural networks [45] have been considered as problematic in many scenarios, *e.g.*, uncertainty estimation of in-distribution samples [19, 24, 30], those of out-of-distribution samples [21, 36, 41], and ensemble learning [35], just to name a few. In the context of adversarial training, Stutz et al. [52] have shown that regularizing confidence on adversarial examples to be uniform can improve detection of adversarial examples from unseen threat models. In this paper, we address the overconfidence at adversarial examples particularly focusing on *smoothed classifiers*, observing that a simple approach of directly fixing the problem could significantly improve the certified robustness.

**Mixup-based training.**   Originally, *mixup* [67] has proposed as a simple yet effective data augmentation scheme to improve generalization and robustness (against small adversarial attacks) of deep neural networks, and there have been significant follow-up works to further improve this form [57, 63, 27, 28]. Recently, Zhang et al. [70] have also explored on theoretical justifications behind how could such an augmentation improves generalization and robustness. Although our method uses a similar linear interpolation scheme of mixup, there is still an essential difference between ours and this line of works: namely, we do not rely on the prior of interpolating two (or more) *independent* samples, but rather aims to directly calibrate predictions between a clean and its (unrestricted) adversarial example, *i.e.*, we consider a new form of *self-mixup* training.

There have been also attempts to employ mixup particularly for improving adversarial robustness: Lamb et al. [31] have shown that an additional mixup loss between adversarial examples upon the standard mixup training achieves a comparable robustness to adversarial training (AT) [40], while not compromising the clean accuracy as much as AT; Lee et al. [37] have proposed *Adversarial Vertex Mixup* to improve AT, by extrapolating predictions along the direction of adversarial perturbation up to few times of its norm via mixup training. Our proposed method can be differentiated to these approaches, in a sense that we employ mixup not to directly improve the robustness of a given neural network, but of its smoothed counterpart. It is also our unique perspective that we consider *unrestricted* adversarial examples to be interpolated.

## E    Variance of results over multiple runs

In our experiments, we report single-run results for ACR and certified robust accuracy as also done by [11, 49, 38, 65, 23], considering that ACR is fairly a robust metric to network initialization: *e.g.*, in Table 5, we report the mean and standard deviation of ACRs across 5 seeds for the MNIST results reported in Table 2. Overall, we confirm that ACR generally shows low variance over multiple runs across a wide range of training methods, including ours.

Table 4: Comparison of certified test accuracy for various training methods on MNIST. The reported values are the mean and standard deviation across 5 seeds. We set our result bold-faced whenever the value improves the Gaussian baseline, and the underlined are best-performing model per $\sigma$.

| $\sigma$ | Models (MNIST) | 0.00 | 0.50 | 1.00 | 1.50 | 2.00 | 2.50 |
|---|---|---|---|---|---|---|---|
| 0.25 | Gaussian [11] | $99.25 \pm 0.04$ | $96.75 \pm 0.11$ | $0.00 \pm 0.00$ | $0.00 \pm 0.00$ | $0.00 \pm 0.00$ | $0.00 \pm 0.00$ |
| | Stability training [38] | $99.34 \pm 0.04$ | $97.12 \pm 0.12$ | $0.00 \pm 0.00$ | $0.00 \pm 0.00$ | $0.00 \pm 0.00$ | $0.00 \pm 0.00$ |
| | SmoothAdv [49] | $99.39 \pm 0.01$ | $98.17 \pm 0.06$ | $0.00 \pm 0.00$ | $0.00 \pm 0.00$ | $0.00 \pm 0.00$ | $0.00 \pm 0.00$ |
| | MACER [65] | $99.33 \pm 0.03$ | $97.35 \pm 0.08$ | $0.00 \pm 0.00$ | $0.00 \pm 0.00$ | $0.00 \pm 0.00$ | $0.00 \pm 0.00$ |
| | Consistency [23] | $99.43 \pm 0.03$ | $97.92 \pm 0.09$ | $0.00 \pm 0.00$ | $0.00 \pm 0.00$ | $0.00 \pm 0.00$ | $0.00 \pm 0.00$ |
| | **SmoothMix** ($\eta = 1.0$) | $\mathbf{99.43 \pm 0.03}$ | $\mathbf{98.10 \pm 0.06}$ | $0.00 \pm 0.00$ | $0.00 \pm 0.00$ | $0.00 \pm 0.00$ | $0.00 \pm 0.00$ |
| | **+ One-Step adversary** | $\mathbf{99.39 \pm 0.02}$ | $\mathbf{98.17 \pm 0.06}$ | $0.00 \pm 0.00$ | $0.00 \pm 0.00$ | $0.00 \pm 0.00$ | $0.00 \pm 0.00$ |
| | **SmoothMix** ($\eta = 5.0$) | $\underline{\mathbf{99.45 \pm 0.03}}$ | $\mathbf{98.17 \pm 0.07}$ | $0.00 \pm 0.00$ | $0.00 \pm 0.00$ | $0.00 \pm 0.00$ | $0.00 \pm 0.00$ |
| | **+ One-Step adversary** | $\mathbf{99.37 \pm 0.02}$ | $\underline{\mathbf{98.20 \pm 0.03}}$ | $0.00 \pm 0.00$ | $0.00 \pm 0.00$ | $0.00 \pm 0.00$ | $0.00 \pm 0.00$ |
| 0.50 | Gaussian [11] | $99.15 \pm 0.03$ | $96.90 \pm 0.06$ | $89.83 \pm 0.06$ | $67.80 \pm 0.16$ | $0.00 \pm 0.00$ | $0.00 \pm 0.00$ |
| | Stability training [38] | $99.26 \pm 0.02$ | $97.27 \pm 0.09$ | $90.75 \pm 0.11$ | $69.15 \pm 0.38$ | $0.00 \pm 0.00$ | $0.00 \pm 0.00$ |
| | SmoothAdv [49] | $99.03 \pm 0.03$ | $97.36 \pm 0.06$ | $92.94 \pm 0.08$ | $81.06 \pm 0.12$ | $0.00 \pm 0.00$ | $0.00 \pm 0.00$ |
| | MACER [65] | $98.69 \pm 0.09$ | $96.28 \pm 0.17$ | $90.14 \pm 0.20$ | $72.12 \pm 0.75$ | $0.00 \pm 0.00$ | $0.00 \pm 0.00$ |
| | Consistency [23] | $99.15 \pm 0.03$ | $97.51 \pm 0.07$ | $92.89 \pm 0.10$ | $78.26 \pm 0.23$ | $0.00 \pm 0.00$ | $0.00 \pm 0.00$ |
| | **SmoothMix** ($\eta = 1.0$) | $99.10 \pm 0.02$ | $\underline{\mathbf{97.51 \pm 0.07}}$ | $\mathbf{92.91 \pm 0.08}$ | $\mathbf{80.15 \pm 0.05}$ | $0.00 \pm 0.00$ | $0.00 \pm 0.00$ |
| | **+ One-Step adversary** | $98.74 \pm 0.04$ | $97.09 \pm 0.06$ | $\mathbf{92.67 \pm 0.05}$ | $\underline{\mathbf{81.70 \pm 0.05}}$ | $0.00 \pm 0.00$ | $0.00 \pm 0.00$ |
| | **SmoothMix** ($\eta = 5.0$) | $98.64 \pm 0.04$ | $96.98 \pm 0.02$ | $92.63 \pm 0.07$ | $\underline{\mathbf{81.85 \pm 0.10}}$ | $0.00 \pm 0.00$ | $0.00 \pm 0.00$ |
| | **+ One-Step adversary** | $98.21 \pm 0.02$ | $96.34 \pm 0.04$ | $91.46 \pm 0.03$ | $\underline{\mathbf{81.20 \pm 0.15}}$ | $0.00 \pm 0.00$ | $0.00 \pm 0.00$ |
| 1.00 | Gaussian [11] | $96.34 \pm 0.03$ | $91.39 \pm 0.05$ | $79.86 \pm 0.08$ | $59.49 \pm 0.10$ | $32.46 \pm 0.20$ | $10.93 \pm 0.12$ |
| | Stability training [38] | $96.43 \pm 0.05$ | $91.63 \pm 0.05$ | $80.45 \pm 0.16$ | $60.53 \pm 0.07$ | $33.35 \pm 0.13$ | $11.05 \pm 0.13$ |
| | SmoothAdv [49] | $95.76 \pm 0.03$ | $90.72 \pm 0.07$ | $80.81 \pm 0.14$ | $64.44 \pm 0.14$ | $43.25 \pm 0.14$ | $22.58 \pm 0.40$ |
| | MACER [65] | $91.59 \pm 0.20$ | $83.44 \pm 0.35$ | $71.10 \pm 0.45$ | $55.67 \pm 0.27$ | $38.67 \pm 0.33$ | $20.09 \pm 0.64$ |
| | Consistency [23] | $94.96 \pm 0.02$ | $89.75 \pm 0.07$ | $79.70 \pm 0.09$ | $63.54 \pm 0.12$ | $41.74 \pm 0.13$ | $20.22 \pm 0.25$ |
| | **SmoothMix** ($\eta = 1.0$) | $95.52 \pm 0.08$ | $90.50 \pm 0.07$ | $\mathbf{80.55 \pm 0.09}$ | $\mathbf{64.09 \pm 0.15}$ | $\mathbf{43.16 \pm 0.05}$ | $\underline{\mathbf{23.94 \pm 0.19}}$ |
| | **+ One-Step adversary** | $94.72 \pm 0.07$ | $89.40 \pm 0.09$ | $79.46 \pm 0.08$ | $\mathbf{64.04 \pm 0.08}$ | $\underline{\mathbf{44.82 \pm 0.09}}$ | $\underline{\mathbf{27.35 \pm 0.15}}$ |
| | **SmoothMix** ($\eta = 5.0$) | $93.71 \pm 0.04$ | $88.00 \pm 0.05$ | $77.95 \pm 0.13$ | $62.78 \pm 0.08$ | $\underline{\mathbf{44.87 \pm 0.14}}$ | $\underline{\mathbf{28.88 \pm 0.16}}$ |
| | **+ One-Step adversary** | $93.11 \pm 0.05$ | $87.24 \pm 0.07$ | $77.22 \pm 0.10$ | $\mathbf{62.48 \pm 0.15}$ | $\underline{\mathbf{44.85 \pm 0.05}}$ | $\underline{\mathbf{29.66 \pm 0.15}}$ |

Table 5: Comparison of ACR for various training methods on MNIST. The reported values are the mean and standard deviation across 5 seeds. We set our result bold-faced whenever the value improves the Gaussian baseline, and the underlined are best-performing model per $\sigma$.

| ACR (MNIST) | $\sigma = 0.25$ | $\sigma = 0.50$ | $\sigma = 1.00$ |
|---|---|---|---|
| Gaussian [11] | $0.9108_{\pm 0.0003}$ | $1.5581_{\pm 0.0016}$ | $1.6184_{\pm 0.0021}$ |
| Stability [38] | $0.9152_{\pm 0.0007}$ | $1.5719_{\pm 0.0028}$ | $1.6341_{\pm 0.0018}$ |
| SmoothAdv [49] | $0.9322_{\pm 0.0005}$ | $1.6872_{\pm 0.0007}$ | $1.7786_{\pm 0.0017}$ |
| MACER [65] | $0.9201_{\pm 0.0006}$ | $1.5899_{\pm 0.0069}$ | $1.5950_{\pm 0.0051}$ |
| Consistency [23] | $0.9279_{\pm 0.0003}$ | $1.6549_{\pm 0.0011}$ | $1.7376_{\pm 0.0017}$ |
| **SmoothMix** ($\eta = 1.0$) | $\mathbf{0.9296}_{\pm 0.0003}$ | $\mathbf{1.6776}_{\pm 0.0007}$ | $\mathbf{1.7867}_{\pm 0.0020}$ |
| **+ One-Step adversary** | $\mathbf{0.9330}_{\pm 0.0004}$ | $\underline{\mathbf{1.6932}}_{\pm 0.0009}$ | $\mathbf{1.8169}_{\pm 0.0011}$ |
| **SmoothMix** ($\eta = 5.0$) | $\mathbf{0.9317}_{\pm 0.0002}$ | $\underline{\mathbf{1.6932}}_{\pm 0.0007}$ | $\mathbf{1.8185}_{\pm 0.0016}$ |
| **+ One-Step adversary** | $\underline{\mathbf{0.9332}}_{\pm 0.0002}$ | $\mathbf{1.6851}_{\pm 0.0003}$ | $\underline{\mathbf{1.8212}}_{\pm 0.0013}$ |

# F   Additional results on CIFAR-10

In this section, we report additional experimental results on CIFAR-10 [29], namely with $\sigma = 1.0$ (see Table 3 for the results for $\sigma \in \{0.25, 0.5\}$). We follow the same experimental details as specified in Section 4.2 and Appendix C, including the common hyperparameter choice of $\eta = 5.0$ for SmoothMix for other experiments as well. Again, we compare our method with various existing robust training methods for smoothed classifiers [11, 38, 49, 65, 23], and Table 6 summarizes the results. Overall, we still observe a similar trend to Section 4.2 that (a) "SmoothMix" offers a significant improvement of robust accuracy without compromising the clean accuracy much, and (b) incorporating the one-step adversary thus can further complementarily boost ACR to outperform other state-of-the-art baseline training methods: *e.g.*, it is notable that "SmoothMix + One-step adversary" achieves fairly comparable or better robust accuracy than MACER while maintaining much higher clean accuracy, *i.e.*, the certified test accuracy at $r = 0.0$, namely $41.4 \rightarrow 45.1$. This confirms that our proposed SmoothMix can offer a better trade-off between accuracy and certified robustness during training.

Table 6: Comparison of approximate certified test accuracy (%) and ACR on CIFAR-10. All the models are trained and evaluated with the same smoothing factor specified by $\sigma$. Each value except ACR indicates the fraction of test samples those have $\ell_2$ certified radius larger than the threshold specified at the top row. We set our results bold-faced whenever the value improves the Gaussian baseline, and underlined whenever the value achieves the best among the considered baselines. * indicates that the results are evaluated from the official pre-trained models released by authors.

| $\sigma$ | Models (CIFAR-10) | ACR | 0.00 | 0.25 | 0.50 | 0.75 | 1.00 | 1.25 | 1.50 | 1.75 | 2.00 | 2.25 |
|---|---|---|---|---|---|---|---|---|---|---|---|---|
| 1.00 | Gaussian [11] | 0.542 | 47.2 | 39.2 | 34.0 | 27.8 | 21.6 | 17.4 | 14.0 | 11.8 | 10.0 | 7.6 |
| | Stability training [38] | 0.526 | 43.5 | 38.9 | 32.8 | 27.0 | 23.1 | 19.1 | 15.4 | 11.3 | 7.8 | 5.7 |
| | SmoothAdv* [49] | 0.660 | 50.8 | 44.9 | 39.0 | 33.6 | 28.5 | 23.7 | 19.4 | 15.4 | 12.0 | 8.7 |
| | MACER* [65] | 0.744 | 41.4 | 38.5 | 35.2 | 32.3 | 29.3 | 26.4 | 23.4 | 20.2 | 17.4 | 14.5 |
| | Consistency [23] | 0.756 | 46.3 | 42.2 | 38.1 | 34.3 | 30.0 | 26.3 | 22.9 | 19.7 | 16.6 | 13.8 |
| | **SmoothMix (Ours)** | **0.725** | 47.1 | **42.5** | **37.5** | **32.9** | **28.7** | **24.9** | **21.3** | **18.3** | **15.5** | **12.6** |
| | **+ One-step adversary** | **0.773** | 45.1 | **41.5** | **37.5** | **33.8** | **30.2** | **26.7** | **23.4** | **20.2** | **17.2** | **14.7** |

# G    Results on ImageNet

We also compare our method on ImageNet [48] classification dataset, to verify the scalability of the method on large-scale datasets. In this experiment, we perform our evaluation on the sub-sampled validation dataset of ImageNet with 500 samples following the previous works [11, 49, 23]. When SmoothMix is used, we simply set $T = 1$ and $m = 1$ mainly in order to reduce the overall training cost, and we fix $\alpha = 8.0$ for both cases of $\sigma = 0.5, 1.0$: this choice leads larger $\alpha \cdot T$ when $\sigma = 0.5$ compared to the MNIST and CIFAR-10 experiments, but we empirically observe that ImageNet is less sensitive to $\alpha \cdot T$, possibly due to that ImageNet consists of higher-resolution inputs, *i.e.*, higher input dimension accordingly, than the others. We use the one-step adversary (Section 3.2) by default here, but we make sure that each adversarial example (found with a large $\alpha$) is further projected in a $\ell_2$-ball of $\epsilon = 1.0$ before it replaces the clean sample, which can be done without adding significant computational overhead. Table 7 summarizes the results, and we still observe the effectiveness of SmoothMix compared to the baseline methods, both in terms of ACR and certified test accuracy.

Table 7: Comparison of approximate certified test accuracy (%) on ImageNet. We set our results bold-faced whenever the value improves the Gaussian baseline, and underlined whenever the value achieves the best among the considered baselines.

| $\sigma$ | Models (ImageNet) | ACR | 0.0 | 0.5 | 1.0 | 1.5 | 2.0 | 2.5 | 3.0 | 3.5 |
|---|---|---|---|---|---|---|---|---|---|---|
| 0.50 | Gaussian [11] | 0.733 | 57 | 46 | 37 | 29 | 0 | 0 | 0 | 0 |
| | Consistency [23] | 0.822 | 55 | 50 | 44 | 34 | 0 | 0 | 0 | 0 |
| | SmoothAdv [49] | 0.825 | 54 | 49 | 43 | 37 | 0 | 0 | 0 | 0 |
| | **SmoothMix (Ours)** | **0.846** | 55 | **50** | 43 | **38** | 0 | 0 | 0 | 0 |
| 1.00 | Gaussian [11] | 0.875 | 44 | 38 | 33 | 26 | 19 | 15 | 12 | 9 |
| | Consistency [23] | 0.982 | 41 | 37 | 32 | 28 | 24 | 21 | 17 | 14 |
| | SmoothAdv [49] | 1.040 | 40 | 37 | 34 | 30 | 27 | 25 | 20 | 15 |
| | **SmoothMix (Ours)** | **1.047** | 40 | 37 | **34** | **30** | **26** | **24** | **20** | **17** |

# H    Detailed results on ablation study

In this section, we report the detailed numerical results and more discussions on the ablation study presented in Section 4.3. Here, Table 8, 9, 10 and 11 presented in what follow are the detailed results for Figure 6, 7(a), 7(b) and 7(c), respectively.

Table 8: Comparison of ACR and approximate certified test accuracy on MNIST for varying $\eta$. We assume $\sigma = 1.0$ in this experiment. "Gaussian" indicates the baseline training with Gaussian augmentation. We set the results bold-faced whenever the value improves "Gaussian".

| Setups | ACR | 0.00 | 0.25 | 0.50 | 0.75 | 1.00 | 1.25 | 1.50 | 1.75 | 2.00 | 2.25 | 2.50 |
|---|---|---|---|---|---|---|---|---|---|---|---|---|
| Gaussian | 1.620 | 96.4 | 94.4 | 91.4 | 87.0 | 79.9 | 71.0 | 59.6 | 46.2 | 32.6 | 19.7 | 10.8 |
| $\eta = 1$ | **1.789** | 95.5 | 93.6 | 90.5 | 86.2 | **80.7** | **73.7** | **64.1** | **53.9** | **43.1** | **33.5** | **24.1** |
| $\eta = 2$ | **1.810** | 94.9 | 92.7 | 89.7 | 85.1 | 79.6 | **72.6** | **63.8** | **54.0** | **44.4** | **35.4** | **26.6** |
| $\eta = 4$ | **1.820** | 94.0 | 91.8 | 88.4 | 83.9 | 78.3 | **71.4** | **63.0** | **53.6** | **44.9** | **36.8** | **28.7** |
| $\eta = 8$ | **1.817** | 93.4 | 91.0 | 87.5 | 82.7 | 77.3 | 70.2 | **62.4** | **53.0** | **44.8** | **37.0** | **29.3** |
| $\eta = 16$ | **1.812** | 92.9 | 90.3 | 86.7 | 82.1 | 76.6 | 69.7 | **61.8** | **52.6** | **44.5** | **36.9** | **29.6** |

Table 9: Comparison of ACR and approximate certified test accuracy on MNIST for varying $\alpha$ and $T$ under control of $\alpha \cdot T = 8$. We assume $\sigma = 1.0$ in this experiment. "Gaussian" indicates the baseline training with Gaussian augmentation. We set the results bold-faced whenever the value improves "Gaussian".

| Setups | ACR | 0.00 | 0.25 | 0.50 | 0.75 | 1.00 | 1.25 | 1.50 | 1.75 | 2.00 | 2.25 | 2.50 |
|---|---|---|---|---|---|---|---|---|---|---|---|---|
| Gaussian | 1.620 | 96.4 | 94.4 | 91.4 | 87.0 | 79.9 | 71.0 | 59.6 | 46.2 | 32.6 | 19.7 | 10.8 |
| $(\alpha, T) = (8.0, 1)$ | **1.785** | 95.5 | 93.5 | 90.5 | 86.0 | **80.5** | **73.1** | **63.9** | **53.5** | **43.3** | **33.2** | **24.0** |
| $(\alpha, T) = (4.0, 2)$ | **1.788** | 95.4 | 93.4 | 90.4 | 85.9 | **80.5** | **73.5** | **63.9** | **53.5** | **43.1** | **33.4** | **24.4** |
| $(\alpha, T) = (2.0, 4)$ | **1.790** | 95.5 | 93.5 | 90.7 | 86.2 | **80.7** | **73.7** | **64.3** | **53.9** | **43.2** | **33.4** | **23.8** |
| $(\alpha, T) = (1.0, 8)$ | **1.789** | 95.5 | 93.6 | 90.5 | 86.2 | **80.7** | **73.7** | **64.1** | **53.9** | **43.1** | **33.5** | **24.1** |

Table 10: Comparison of ACR and approximate certified test accuracy on MNIST for varying $\varepsilon$, the hard limit on $\ell_2$-norm of adversarial perturbations. We assume $\sigma = 1.0$ in this experiment. "Gaussian" indicates the baseline training with Gaussian augmentation. "$\varepsilon = \infty$" denotes our original setup of unrestricted adversarial attacks. We set the results bold-faced whenever the value improves "Gaussian".

| Setups | ACR | 0.00 | 0.25 | 0.50 | 0.75 | 1.00 | 1.25 | 1.50 | 1.75 | 2.00 | 2.25 | 2.50 |
|---|---|---|---|---|---|---|---|---|---|---|---|---|
| Gaussian | 1.620 | 96.4 | 94.4 | 91.4 | 87.0 | 79.9 | 71.0 | 59.6 | 46.2 | 32.6 | 19.7 | 10.8 |
| $\varepsilon = 2.0$ | **1.723** | 96.1 | 94.3 | 91.4 | **87.1** | **81.2** | **73.6** | **63.7** | **52.1** | **39.8** | **28.2** | **16.6** |
| $\varepsilon = 4.0$ | **1.751** | 95.9 | 94.0 | 91.1 | 86.8 | **81.0** | **73.7** | **64.3** | **53.1** | **41.4** | **30.6** | **19.8** |
| $\varepsilon = 6.0$ | **1.778** | 95.6 | 93.7 | 90.6 | 86.5 | **80.8** | **73.7** | **64.4** | **53.8** | **42.8** | **32.6** | **22.9** |
| $\varepsilon = 8.0$ | **1.788** | 95.5 | 93.5 | 90.4 | 86.1 | **80.5** | **73.5** | **64.2** | **53.8** | **43.2** | **33.5** | **24.1** |
| $\varepsilon = \infty$ (Ours) | **1.789** | 95.5 | 93.6 | 90.5 | 86.2 | **80.7** | **73.7** | **64.1** | **53.9** | **43.1** | **33.5** | **24.1** |

Table 11: Comparison of ACR and approximate certified test accuracy on MNIST for varying $m$, the number of noise samples used for estimating smoothed predictions. We assume $\sigma = 1.0$ in this experiment. "Gaussian" indicates the baseline training with Gaussian augmentation. We set the results bold-faced whenever the value improves "Gaussian".

| Setups | ACR | 0.00 | 0.25 | 0.50 | 0.75 | 1.00 | 1.25 | 1.50 | 1.75 | 2.00 | 2.25 | 2.50 |
|---|---|---|---|---|---|---|---|---|---|---|---|---|
| Gaussian | 1.620 | 96.4 | 94.4 | 91.4 | 87.0 | 79.9 | 71.0 | 59.6 | 46.2 | 32.6 | 19.7 | 10.8 |
| $m = 1$ | **1.744** | 94.5 | 92.2 | 88.9 | 84.1 | 78.1 | 70.9 | **61.9** | **51.7** | **41.7** | **31.9** | **23.2** |
| $m = 2$ | **1.776** | 95.3 | 93.0 | 89.8 | 85.4 | 79.8 | **72.7** | **63.5** | **53.1** | **42.6** | **33.0** | **24.0** |
| $m = 4$ | **1.789** | 95.5 | 93.6 | 90.5 | 86.2 | **80.7** | **73.7** | **64.1** | **53.9** | **43.1** | **33.5** | **24.1** |
| $m = 8$ | **1.788** | 95.9 | 93.9 | 91.0 | 86.7 | **81.0** | **73.9** | **64.6** | **54.1** | **43.2** | **33.1** | **23.3** |