# OpenReview forum: "SmoothMix: Training Confidence-calibrated Smoothed Classifiers for Certified Robustness"
_NeurIPS.cc/2021/Conference — NeurIPS 2021 Poster_

### Official Review · Reviewer_8Cj1 · 2021-07-04

**Rating:** 6
**Confidence:** 4

**Summary:**

The paper offers a novel training scheme—called SmoothMix—for fitting deep net classifiers with improved adversarial robustness while maintaining accuracy to the extent possible. The key idea is to combine the mixup loss with smooth classifiers. The former interpolates the input sample with its adversarial perturbation, designed to “mislead” the smoothed classifier. The effectiveness of the proposed idea was studied on MNIST, CIFAR10, and ImageNet, showing competitive performance to that of state-of-the-art adversarial defense strategies (Gaussian smoothing, stability training, adversarial smoothing, MACER, and consistency).

**Limitations And Societal Impact:**

It will be a plus to explain that the SmoothMix approach has an additional hyper-parameter to tune (\alpha).

**Main Review:**

Originality

The idea of combining the mixup loss with adversarial smoothing is novel. Also, it is nicely motivated. First, it was demonstrated (visually) that adversarial perturbations of a smoothed classifier may have semantic changes that either transform the input to another class or removing relevant content for the current class. Second, the smoothed classifier tends to assign to adversarially-crafted examples higher confidence scores compared to the clean input.

Motivated by the above observations the authors hypothesized that miscalibration (although not defined precisely) degrades the certified robustness of smoothed classifiers. They propose to penalize over-confident predictions by interpolating the original image (and label) with its unrestricted adversarial example, assigning uniform confidence to the latter. This approach encourages the smoothed classifier to produce a uniform confidence score for adversarial samples.

Quality

The paper is well written, the exposition of the idea is clear, the experiments follow existing literature.

Main concern

While the method attains state-of-the-art performance, it is hard to judge whether the new results are significantly better than existing ones, e.g., they are only slightly better than adversarial smoothing. It will be interesting to see a detailed performance analysis, e.g., a report of class-conditional performance metrics could perhaps communicate better the advantages of the proposed method.

Significance

Certified adversarial robustness is an active area of research that sheds light on the robustness of deep nets. The proposed method achieves state-of-the-art performance on benchmark datasets. It is likely that---at least in the short term---this new method will have a noticeable impact.

**Time Spent Reviewing:**

6

---

> ### Author Response · Authors · 2021-08-10
> **Official response to Reviewer 8Cj1**
>
> We sincerely appreciate your thoughtful comments, efforts, and time. We respond to each of your questions and concerns one-by-one in what follows. We also kindly ask you to check out the *common response* we have posted together, which addresses your concern on the significance of our empirical results. All the responses will be carefully incorporated in the final draft.
>
> ---
> **Q1. Additional hyperparameter $\alpha$?**
>
> Thanks for your suggestion. Nevertheless, we note that our method and SmoothAdv use the same number of hyperparameters: although SmoothMix does introduce $\alpha$ (compared to SmoothAdv), it instead does not use the hyperparameter $\varepsilon$ of SmoothAdv, i.e., the maximum norm of adversarial perturbations. We will clarify this in the final draft.

---

> > ### Comment · Reviewer_8Cj1 · 2021-08-26
> > **Follow-up**
> >
> > Thank you for your response and clarifications.

---

> > > ### Author Response · Authors · 2021-08-30
> > > **Thank you for the response**
> > >
> > > Thank you again for the acknowledgement on our rebuttal.
> > >
> > > If you have any remaining suggestions or concerns, please let us know!
> > >
> > > Best, Authors.

---

### Official Review · Reviewer_jNUT · 2021-07-11

**Rating:** 6
**Confidence:** 4

**Summary:**

The paper observes a connection between the confidence with which a smooth model classifies an input and the robustness radius with which the input's prediction can be certified. In particular, the paper argues that the model's over-confidence on some inputs can be a source for detrimental effects on the certified radius of other (nearby) instances. Inspired by this, the paper proposes to find over-confident near off-class samples (somewhat) near training instances, and conduct mixup training on this pair of instances as a way of improving the certified accuracy of the resulting smoothed classifier. The proposal can be regarded as another way of adversarial training that is better crafted for smoothed classifiers. Experiments conducted on MNIST, CIFAR and ImageNet show marginal but consistent improvements in certified accuracy.

**Limitations And Societal Impact:**

Yes, they have addressed it.

**Main Review:**

Strengths:
+ The paper's object of study, i.e. modifying the training of smooth classifiers for enhancing the classifier's certified accuracy, is of interest to the adversarial robustness and certification communities, and in someway of interest for the deep learning community in general.
+ The paper's writing is mostly clear, following a narrative that properly exposes the paper's motivation, relation to previous works and the rationale behind the experiments that are conducted. Figures are also well-built and clear.
+ The experimental protocol is standard and sensible.
+ Experimental results are mostly consistent across factors being studied (datasets, parameters, etc.).
+ The paper's motivation and formulation (although as I expose next I do not entirely understand or agree with) draws on previous works but is for the largest part original.

Weaknesses that affect my rating:
+ In general I find two main issues with the paper: (1) the motivation is still not completely clear to me, and (2) the experimental results are somewhat not significant enough. In particular, I find the motivation rather troubling, as I am not entirely sure about the over-confidence item raised in the paper: is this over-confidence truly a common phenomenon in *smoothed* classifiers? Given the relation between smoothed classifiers and "Lipschitzness" I would have initially thought of the opposite, and so I think this (fundamental) claim in the paper would require experimental validation. Further, while the paper repeats several times that the adversaries are "unconstrained" in practice that does not appear to be the case, precisely because part of the importance of finding adversarial examples is that they are near the original inputs, so perhaps a more precise term for this would simply be adversarial examples with looser constraint.

Next, I outline some questions that better expose what I find unclear about the paper, and so I would be thankful for further explanation.
+ Eqn. (8) implies that the adversarial examples that the optimization procedure most likely finds are the ones for which the label $y$ has low score. How is this quantity related to "over-confidence", could it not be the case that simply the classifier will be not confident towards $y$ but uniformly confident towards the rest of the classes? Why would over-confidence emerge here?
+ I do not understand the claims made in the bullet point of L138. Fig. 2 compares apparently two cases: (a) when the input suffers some modifications so that the label changes and (b) when the modifications to the input can be simply described as removing characteristic features of the class being described (the deer's antlers, in this case). How are these two cases related to how sensible it is for the classifier to keep its low confidence to the original class? Is the paper stating that the classifier's confidence for the true class when making predictions over the original input should be low because it should be aware of how few changes would be required to change the semantic class? I do not think I would agree with such a premise: even though zebras and horses look very much alike, when perceiving a zebra we can be rather certain it is a zebra and not a horse, precisely because of its *very* distinctive features.
+ In general, while the experimental results are mostly consistent, the improvements are rather marginal, and so I am unsure whether they are significant.
+ My understanding of one of this work's main contributions is observing a relation between the certification of a classifier and its calibration. Because of this observation, the paper proposes to improve calibration through a mixup-inspired procedure so that certification is also improved. Is my understanding correct? If so, would not a valid baseline be other previous calibration techniques? Or is there some issue with this approach? If there is no issue, how do other calibration techniques perform towards improving certification?


Weaknesses that do not affect my rating but should be addressed if possible:
+ L6: of smoothed classifier
+ L8: it trains convex combinations? or *on* convex combinations?
+ In Fig. 1's caption: "mixup" or "the mixup"?
+ L37: one of important
+ L57: they goes
+ L85: construct
+ L142: "their" or "its"?
+ L144: have
+ L146: "a semantically off-class samples those to be labeled as the uniform confidence"
+ L151: those
+ Fig. 2's caption is not self-contained: what the authors want to express with the figure is unclear from the caption. In particular, what is meant by "in/out-of-class translation"? I think this caption requires more explanation.
+ In L159, I do not understand what the paper means by "mixing the uniform confidence to them". What does "uniform confidence" mean, and to whom is it being "mixed"? Similarly, in L166 the paper says "direct supervision of the uniform confidence on it", which I do not understand. Could the authors please clarify what was meant by these expressions?
+ Table 1's caption: "those"
+ The tables reporting the experiments follow a protocol of adding bold-face and underline that I think is confusing: usually whenever readers see a bold-faced number (I think) they think it is the largest, which is not the case in this paper. I would urge the authors to reconsider this protocol. (For instance the results of other methods are never underlined, which can misguide th reading of results)
+ Table 1's caption: "improves" or "outperforms"?
+ Fig. 4's caption: "plots"
+ L195: trainig
+ Fig. 5's caption: "plots"
+ L230: improve
+ L240: those
+ For the experiment reported in L285, how were the adversarial examples computed? Were both restricted or both unrestricted? Or was there a mismatch in this? If there was a mismatch, I think the results are not interpretable, as the process of finding an adversary for the Gaussian model is constrained and so biased towards small values, while the one for SmoothMix is not.
+ I encourage the authors to rename this method, as there is a paper with the same name: "SmoothMix: a Simple Yet Effective Data Augmentation to Train Robust Classifiers", Lee et al., CVPR20


**Time Spent Reviewing:**

6

---

> ### Author Response · Authors · 2021-08-10
> **Official response to Reviewer jNUT**
>
> We sincerely appreciate your thoughtful comments, efforts, and time. We respond to each of your questions and concerns one-by-one in what follows. We also kindly ask you to check out the *common response* we have posted together, which addresses your concern on the significance of our empirical results. All the responses will be carefully incorporated in the final draft.
>
> ---
> **Q1. How overconfidence happens in smoothed classifiers of Lipschitzness?**
>
> Thanks for asking an important question. As you mentioned, smoothed classifiers are naturally Lipschitz, therefore it is unlikely that one can find an overconfident adversarial example inside a small-norm ball around a given input, e.g., an $\ell_2$-ball of size $\varepsilon=0.5$ commonly used in the literature of adversarial training [Madry et al., 2018]. Here we emphasize that, however, our focus is *far beyond* this restriction: e.g., we even allow the adversarial examples to have $\varepsilon$ up to 5~10 during training. Our key finding is that even smoothed classifiers reveal “overconfident” examples in this scenario, i.e., their confidences are abnormally higher than those of clean samples, while the size of perturbations are mostly not-too-large to completely change the semantic of the original input. The table below indeed supports this phenomenon, in case of CIFAR-10 with ResNet-110 assuming $\sigma=0.5$: here, we can observe that the off-class confidence $\mathbb{E}[\max_{c\neq y}\hat{f}_c(x)]$ could exceed the maximum clean confidence $\mathbb{E}[\hat{f}_y(x)]$ as we allow more budget on the perturbation size. We will further clarify this point and include the respective discussion in the final draft.
>
> | CIFAR-10  |  Clean  |  $\varepsilon$ = 1 | $\varepsilon$ = 2 | $\varepsilon$ = 3 | $\varepsilon$ = 4 | $\varepsilon$ = 5 |
> |--------:|:------------:|:----:|:----:|:--------------:|:----:|:----:|
> |  $\mathbb{E}[\hat{f}_y(x)]$ (%)  |  **66.4**  | 47.1  | 24.3 | 14.2  | 11.3  | 10.7 |
> |  $\mathbb{E}[\max_{c\neq y}\hat{f}_c(x)]$ (%) |  24.2  | 37.8  | 59.5 |  **71.8**  | **78.5** | **82.0** |
>
> ---
> **Q2. The term “unconstrained” may not be precise.**
>
> Thanks for your suggestion, and we agree that the term may confuse the readers a bit in a sense that the found adversarial examples should still be correlated to the given input (as described in Line 128-129 and Eq (7)). We will carefully revise our final draft to better convey the concept as per your suggestion.
>
> ---
> **Q3. Eq (8): Low confidence at $y$ may not imply over-confidence?**
>
> As you pointed out, adversarial examples (AEs) found via Eq (8) do not necessarily have to be overconfident towards a particular class, and can get a uniformly-distributed confidence instead. An important point here is that, however, our motivation and the loss design (Eq (10)) *do not* assume that the found AEs must be overconfident. Rather, our focus is to develop a regularization term that acts only if there exists an overconfident AE nearby a given input: more concretely, our proposed cross-entropy loss (Eq (10)) would assign relatively low loss values in cases when the found AE already gets a prediction close to the uniform. We will clarify this point in the final draft.
>
> ---
> **Q4. Line 138: Does it mean that the true-class confidence of the original input should be low?**
>
> No. We point out that our loss (Eq (10)) always labels a clean (original) input, say $x$, to $\hat{F}(x)$, i.e., the current prediction of itself. Hence, the mixup loss itself would not particularly affect the confidences of clean inputs, and they are instead controlled by the natural loss $L^{\tt nat}$ in Eq (11). The comments in Line 138 rather discuss about the confidences at the *adversarial* examples, e.g., those illustrated in Figure 2.
>
> ---
> **Q5. Existing calibration techniques as a baseline?**
>
> An immediate issue to apply existing calibration techniques for smoothed classifiers is that the actual confidence of our interest, i.e., $p_f(x) := \max_c  \mathbb{P}_{\delta} (f(x+\delta)=c)$ (Eq (3)), is quite non-trivial to control compared to those from standard neural networks: e.g., a common calibration technique of temperature scaling [Hinton et al., 2015] of $f$ cannot change the value of $p_f(x)$. Another aspect we note is that our focus in this paper itself, i.e., calibration of distant adversarial examples, has not been addressed even in the literature of confidence calibration. As also mentioned in Section 5, nevertheless, we agree that bridging the techniques in uncertainty estimation for the robustness of smoothed classifiers would be a promising future direction, and will add respective discussion to the final draft.
>
> ---
> **Q6. Line 285: How were the adversarial examples computed? Was there a mismatch in it between the two models?**
>
> There is no mismatch in the attack details to find the adversarial examples from the two classifiers: in the experiment in Line 285 (i.e., a comparison of Equal-confidence mixing ratios), we find adversarial examples from the CIFAR-10 test samples separately from two pre-trained (smoothed) classifiers each trained by (a) Gaussian (Eq (4)), and (b) SmoothMix (Eq (11); ours), respectively. We optimize each adversarial example assuming only a quite loose norm-bound of $\varepsilon=8$ to allow more update steps, i.e., via 50-step PGD for both classifiers (a) and (b). We will further clarify these experimental details in the final draft.
>
> ---
> **Q7. Other minor or editorial comments.**
>
> Many thanks for the careful reading, and making constructive suggestions to improve the clarity of our manuscript. We will gratefully incorporate all your minor or editorial comments in the final draft.
>
> ---
> - [Madry et al., 2018] Towards Deep Learning Models Resistant to Adversarial Attacks, ICLR 2018.
> - [Hinton et al., 2015] Distilling the Knowledge in a Neural Network, 2015.

---

> > ### Comment · Reviewer_jNUT · 2021-08-29
> > **Answer to author's response**
> >
> > Thank you for the follow-up answers, clarifications and experiments. I think most of my concerns have been adequately addressed. In particular, I now understand more about the paper's motivation, although I must also state that yet some of the claims could still be up for discussion (as other reviewers also pointed out their somewhat speculative nature).
> >
> > Given this, I still consider this paper should be accepted, and so I will keep my rating.

---

> > > ### Author Response · Authors · 2021-08-30
> > > **Thank you for the response**
> > >
> > > We are happy to hear that our rebuttal addressed your concerns well.
> > >
> > > Thank you again for the valuable suggestions and comments, which we will incorporate in the final version to further strengthen our paper.
> > >
> > > If you have any remaining suggestions or concerns, please let us know!
> > >
> > > Best, Authors.

---

### Official Review · Reviewer_DMPv · 2021-07-15

**Rating:** 6
**Confidence:** 4

**Summary:**

This paper looks to improve upon SmoothAdv - a methodology that employs adversarial training on a smoothed classifier to further improve certified robustness. The methodology employed looks to tackling overconfident examples rather than adversarial examples by utilising MixUp. They demonstrate that we can combine MixUp with SmoothAdv to improve certified accuracy.

**Limitations And Societal Impact:**

Yes

**Main Review:**

UPDATE:
The authors have made clearer the motivation of overconfident examples, as a result I am increasing my score to 6.

Strengths:
- This paper demonstrates we can combine MixUp and adversarial training to improve certified accuracy for smoothed classifiers.

Weakness and Questions:
- One main concern I have is that the improvements shown in Tables 1 and 2 seems marginal, for example SmoothMix obtains 55.3% accuracy for CIFAR-10 when $\sigma$=0.25 whereas for Consistency it is 55.2%. Given these small margins, the authors should also report error bars on the approximate certified accuracy.

- The main motivation regarding why overconfident examples should be penalised is talked about in Section 3.1. I found this section a bit confusing to read, in particular the two bullet points seems speculative.  I've read bullet point one several times and I'm still unclear what it is trying to say. Further, both bullet points refers to Figure 2 which pinpoints to just two examples. To ensure these are not cherry-picked or anecdotal examples, the authors should show statistically that smoothed classifiers is overconfident towards the direction of adversarial perturbation.

- It seems throughout when SmoothAdv was used in conjunction, the attack used was only one-step. This has been to known to cause gradient obfuscation / catastrophic overfitting especially when the perturbation radius is large, given that the adversarial example here is unrestricted it would be good to see the results for multi-step attacks as well. Otherwise it is unclear what the one-step attack is doing.

- From Figure 5/6 it would appear that SmoothMix performs worse when the certification radius is smaller compared to SmoothMix and Consistency. This doesn’t seem desirable, can the authors explained why this is?


**Time Spent Reviewing:**

2

---

> ### Author Response · Authors · 2021-08-10
> **Official response to Reviewer DMPv**
>
> We sincerely appreciate your thoughtful comments, efforts, and time. We respond to each of your questions and concerns one-by-one in what follows. We also kindly ask you to check out the *common response* we have posted together, which addresses your concern on the significance of our empirical results. All the responses will be carefully incorporated in the final draft.
>
> ---
> **Q1. Table 2: The ACR gap in "SmoothMix vs. Consistency" is marginal at $\sigma=0.25$?**
>
> We remark that the superiority of our method (compared to baselines) is sometimes better visible from the *certified accuracies* than ACR. For example, in the case of “Consistency vs. SmoothMix” highlighted in the table below (CIFAR-10 at $\sigma=0.25$; taken from Table 2), as concerned by you, one can still check that SmoothMix can take a better trade-off by comparing the certified accuracy $r=0.0$ (or the clean accuracy), i.e., 75.8% (Consistency) vs. 77.1% (SmoothMix), even if they achieve competitive ACR and the certified accuracy at $r=0.75$. This is because an increase in the clean accuracy (i.e., the smoothed classifier correctly classifies more test samples with CR > 0) often contributes less to increase the ACR value. We will incorporate the respective discussion in the final draft.
>
> | Method  |   ACR   | 0.00 | 0.25 | 0.50 | 0.75 |
> |:--------|:------------:|:----:|:----:|:-----:|:-----:|
> | Gaussian  | 0.424 | 76.6 | 61.2 | 42.2 | 25.1 |
> | Consistency   | 0.552 | 75.8 | 67.6 | **58.1** | **46.7** |
> | SmoothMix (Ours)    | **0.553** | **77.1** | **67.9** | 57.9 | **46.7** |
>
>
> ---
> **Q2. Error bars.**
>
> For your information, we note that we have discussed in Appendix F about the variance of our results across multiple runs, also reporting error bars for ACRs in Table 1. The table below extends this at $\sigma=0.5$ to include error bars for the certified accuracies as well, following your suggestion. Overall, we observe that both ACR and certified accuracy show quite low variances across different seeds. We agree that adding error bars would improve the presentation of our results, and we will incorporate these and further results in the final draft, as well as the error bars for the CIFAR-10 results (i.e., Table 2).
>
> | Method | ACR | 0.00 | 0.25 | 0.50 | 0.75 | 1.00 | 1.25 | 1.50 | 1.75 |
> |:----------|:----------:|:----------:|:----------:|:----------:|:----------:|:----------:|:----------:|:----------:|:----------:|
> | Gaussian | 1.5581 $\pm$ 0.0016 | 99.15 $\pm$ 0.03 | 98.33 $\pm$ 0.04 | 96.90 $\pm$ 0.06 | 94.31 $\pm$ 0.08 | 89.83 $\pm$ 0.06 | 81.98 $\pm$ 0.24 | 67.80 $\pm$ 0.16 | 44.95 $\pm$ 0.32 |
> | Stability training | 1.5719 $\pm$ 0.0028 | 99.26 $\pm$ 0.02 | 98.52 $\pm$ 0.02 | 97.27 $\pm$ 0.09 | 94.91 $\pm$ 0.07 | 90.75 $\pm$ 0.11 | 83.11 $\pm$ 0.15 | 69.15 $\pm$ 0.38 | 45.77 $\pm$ 0.56 |
> | SmoothAdv | 1.6872 $\pm$ 0.0007 | 99.03 $\pm$ 0.03 | 98.37 $\pm$ 0.03 | 97.36 $\pm$ 0.06 | 95.67 $\pm$ 0.06 | 92.94 $\pm$ 0.08 | 88.46 $\pm$ 0.07 | 81.06 $\pm$ 0.12 | 67.48 $\pm$ 0.16 |
> | MACER | 1.5899 $\pm$ 0.0069 | 98.69 $\pm$ 0.09 | 97.79 $\pm$ 0.09 | 96.28 $\pm$ 0.17 | 93.92 $\pm$ 0.10 | 90.14 $\pm$ 0.20 | 83.53 $\pm$ 0.41 | 72.12 $\pm$ 0.75 | 52.33 $\pm$ 0.87 |
> | Consistency | 1.6549 $\pm$ 0.0012 | 99.15 $\pm$ 0.03 | 98.52 $\pm$ 0.07 | 97.51 $\pm$ 0.07 | 95.80 $\pm$ 0.05 | 92.89 $\pm$ 0.10 | 87.68 $\pm$ 0.11 | 78.26 $\pm$ 0.23 | 60.44 $\pm$ 0.20 |
> | **SmoothMix (Ours)** | 1.6932 $\pm$ 0.0007 | 98.64 $\pm$ 0.04 | 97.96 $\pm$ 0.06 | 96.98 $\pm$ 0.02 | 95.20 $\pm$ 0.03 | 92.63 $\pm$ 0.07 | 88.45 $\pm$ 0.05 | 81.85 $\pm$ 0.10 | 69.82 $\pm$ 0.20 |
>
>
> ---
> **Q3. Clarity of Section 3.1 (and Figure 2).**
>
> As per your suggestion, we will revise our manuscript to further improve the clarity of Section 3. We highlight here, for instance, our key motivation on the existence of overconfident examples by comparing the average true-class confidence ($\mathbb{E}[\hat{f}\_{y}(x)]$) and their off-class confidence ($\mathbb{E}[\max\_{c\neq y}\hat{f}_c(x)]$) of a smoothed classifier trained on CIFAR-10 with ResNet-110 and $\sigma=0.5$. The table below confirms that the off-class confidence of adversarial examples can be abnormally higher than those of clean samples, while the size of perturbations are mostly not-too-large to completely change the semantic of the original input. To the best of our knowledge, such an overconfidence issue of “distant” adversarial examples has not been focused on in the prior works, and we show that a method that directly fixes this overconfidence could be a better design of adversarial training. We will include a more detailed analysis in the final draft to better convey our motivation to the readers.
>
> | CIFAR-10  |  Clean  |  $\varepsilon$ = 1 | $\varepsilon$ = 2 | $\varepsilon$ = 3 | $\varepsilon$ = 4 | $\varepsilon$ = 5 |
> |--------:|:------------:|:----:|:----:|:--------------:|:----:|:----:|
> |  $\mathbb{E}[\hat{f}_y(x)]$ (%)  |  **66.4**  | 47.1  | 24.3 | 14.2  | 11.3  | 10.7 |
> |  $\mathbb{E}[\max_{c\neq y}\hat{f}_c(x)]$ (%) |  24.2  | 37.8  | 59.5 |  **71.8**  | **78.5** | **82.0** |
>
> ---
> **Q4. The one-step adversary may cause gradient obfuscation or catastrophic overfitting?**
>
> It is unlikely that our proposed one-step adversary causes either gradient obfuscation (i.e., a false increase in the robust accuracy) or catastrophic overfitting (i.e., a decrease in the test robust accuracy), as our experimental results in Table 1 and 2 already confirm its effectiveness to improve the *provable* guarantee on the test robust accuracy. Although we did consider its “multi-step” variant by performing a few-steps of SmoothAdv before applying SmoothMix, we found the gain from this variant is quite marginal (as mentioned in Line 176). The one-step adversary is in an attempt to also utilize the intermediate adversarial examples found during the multi-step optimization of Eq (8) in SmoothMix, where we found that taking the first-step inputs can be an effective approximation of SmoothAdv. We also remark that the attack objective of our method (Eq (8); and SmoothAdv as well) adds Gaussian noise to inputs in order to directly attack the smoothed classifier instead of the base classifier: this can be another explanation on why the one-step attack does not suffer from gradient obfuscation. We will further clarify this point in the final draft.
>
> ---
> **Q5. Why does SmoothMix often get less certified accuracy than baselines at small radii?**
>
> The decreased robust accuracy at small radii (including the clean accuracy at $r=0.0$ as well) could occur due to the fundamental trade-off between accuracy and robustness [Zhang et al., 2019]: increasing the robust radii of some samples can be at expense of the decreased radii of other (harder) samples. For example, one can check from Table 2 that most robust training baselines also suffer from this degradation in the certified accuracy at $r=0.0$ compared to Gaussian. Therefore, a more tangible objective for smoothed classifiers would be to better balance between the accuracy drop (at small radii) and the robustness (at large radii): a current workaround in the literature is to evaluate the average certified radius (ACR; Line 212 for the details) [Zhai et al., 2020; Jeong and Shin, 2020], as it naturally assigns 0 for the incorrectly classified samples, while giving more weights for samples the classifier can robustly classify. This is one reason why we chose ACR as the major performance metric in this paper.
>
> ---
> - [Zhang et al., 2019] Theoretically Principled Trade-off between Robustness and Accuracy, ICML 2019.
> - [Zhai et al., 2020] MACER: Attack-free and Scalable Robust Training via Maximizing Certified Radius, ICLR 2020.
> - [Jeong and Shin, 2020] Consistency Regularization for Certified Robustness of Smoothed Classifiers, NeurIPS 2020.

---

> > ### Author Response · Authors · 2021-08-24
> > **A gentle reminder for Reviewer DMPv**
> >
> > Dear Reviewer DMPv,
> >
> > Thank you very much again for your time and efforts in reviewing our paper.
> >
> > We kindly remind that we have only a week or so in the discussion period.
> >
> > We wonder whether there is any further concern and hope to have a chance to respond before the discussion phase ends.
> >
> > Regards,
> >
> > Authors

---

### Official Review · Reviewer_cLLo · 2021-07-15

**Rating:** 6
**Confidence:** 4

**Summary:**

The paper proposes a new method for training models to achieve better certified L-2 robustness. To this extent, SmoothMix generates adversarial examples by relaxing the perturbation budget and minimizes a mixup loss. The proposed procedure identifies instances classified with high confidence and near-off-class samples as causes of limited robustness in smoothed classifiers and offers an intuitive way to adaptively set a new decision boundary between these samples for better robustness. Finally, the method obtains improved certified accuracy for certain radii ε that each adversarial perturbation must be in.

**Limitations And Societal Impact:**

Yes, the authors have clearly detailed the limitations and the broader societal impact of the work.

**Main Review:**

Strengths
1. The proposed method is motivated and well-explained.
2. Evaluation results using standard datasets, such as MNIST (Table 1),  Cifar-10 (Table 2), and ImageNet (Table 4), show the effectiveness of SmoothMix across different smoothing factors.
3. Extensive ablation studies aid in understanding the utility of SmoothMix.

Open concerns
1. Average certified radius (ACR) is used to quantify the effectiveness of SmoothMix. However, it may be sensitive to particular adversarial samples, i.e., a few generated adversarial instances could have a high ACR score and thus can mislead the global interpretation of ACR. Analyzing the distribution of ACR scores might shed more light on this point.
2. SmoothMixs' performance on the certified accuracy at different radii and epsilons is not consistent across datasets. In addition, it achieves marginal improvement in most cases. This is observed even for easier datasets like MNIST, where we see the difference gap getting bigger for higher epsilons (this is also arguable as to how practical it is to generate adversarial samples using such larger perturbations).

**Time Spent Reviewing:**

7

---

> ### Author Response · Authors · 2021-08-10
> **Official response to Reviewer cLLo**
>
> We sincerely appreciate your thoughtful comments, efforts, and time. We respond to each of your questions and concerns one-by-one in what follows. We also kindly ask you to check out the *common response* we have posted together, which addresses your concern on the significance of our empirical results. All the responses will be carefully incorporated in the final draft.
>
> ---
> **Q1. ACR may be sensitive to few samples of high ACR scores.**
>
> Although the *average certified radius* (ACR) [Zhai et al., 2020] may not be the best metric to evaluate the certified robustness, we report ACR because (a) it is yet the most commonly-used metric in the literature, and (b) we think ACR still can be a useful one to effectively quantify the trade-off between accuracy and robustness of a smoothed classifier. For (b), we remark that ACR naturally assigns 0 for the incorrectly classified samples, while giving more weights for samples the classifier can robustly classify. One may concern the low breakdown point of sample mean estimators that ACR is based on, e.g., as you mentioned, but in practice each of the certified radii (CRs) becomes strictly upper-bounded to a certain value during the CERTIFY procedure, hence there would be no single sample that has an arbitrarily large CR: e.g., at $\sigma=0.25$, the standard CERTIFY procedure cannot return a CR that exceeds $\approx$ 0.953. This is also visualized in Figure 4 and 5, and one can also check from these plots on how the certified radii are actually distributed in more detail.
>
> We additionally note that the *certified test accuracy* we also report in the paper can supplement ACR: for example, the results highlighted in the below table (CIFAR-10 at $\sigma=0.25$; taken from Table 2) confirms the superiority of our method (SmoothMix) compared to Consistency by the certified accuracy $r=0.0$ (i.e., the clean accuracy), even if they achieve competitive ACR and the certified accuracy at $r=0.75$.
>
> | Method  |   ACR   | 0.00 | 0.25 | 0.50 | 0.75 |
> |:--------|:------------:|:----:|:----:|:-----:|:-----:|
> | Gaussian  | 0.424 | 76.6 | 61.2 | 42.2 | 25.1 |
> | Consistency   | 0.552 | 75.8 | 67.6 | **58.1** | **46.7** |
> | SmoothMix (Ours)	    | **0.553** | **77.1** | **67.9** | 57.9 | **46.7** |
>
> Nevertheless, we agree that developing a more sensible evaluation metric for smoothed classifiers would be an interesting future work. We will add the respective discussions above in the final draft.
>
> ---
> **Q2. The accuracy gains are not consistent across different radii?**
>
> As also discussed in the *common response*, our experimental results are consistent in many aspects, e.g., across datasets, $\sigma$, and random seeds. However, it could be practically hard to be consistent across different radii of robust accuracy, due to the fundamental trade-off between accuracy and robustness [Zhang et al., 2019]: increasing the robustness at some samples can be at expense of an unexpected decrease in accuracy at small radii (including the clean accuracy at $r=0.0$ as well), as also observed in other robust training baselines compared to Gaussian. Therefore, a more tangible objective would be to better balance between the accuracy drop (at small radii) and the robustness (at large radii). Our experimental results indeed focus on this point, showing either (a) an improved ACR compared to the prior arts, or (b) an improved overall certified accuracy in cases when our model is directly comparable to baselines even in the certified accuracy. Ultimately, we believe that closing the accuracy gap at small radii while maintaining the high robustness would be an important step to overcome the current practice of assuming small and fixed threat models in the literature of adversarial robustness. We will clarify this point in the final draft.
>
> ---
> - [Zhai et al., 2020] MACER: Attack-free and Scalable Robust Training via Maximizing Certified Radius, ICLR 2020.
> - [Zhang et al., 2019] Theoretically Principled Trade-off between Robustness and Accuracy, ICML 2019.

---

> > ### Comment · Reviewer_cLLo · 2021-08-13
> > **Discussion response**
> >
> > I appreciate the authors for providing a detailed response. See below for my remaining concerns.
> >
> > **Re Q1.** I get the point of using average certified radius (ACR) as it is the most commonly-used metric in the literature, but that does not justify the disadvantages of the metric. I believe the main argument is whether these disadvantages lead to a false interpretation of results and how practical is the incorporation of distant adversarial examples during training. Further, the author's response of "samples with zero ACR scores are unutilized in the analysis" is unclear as the definition of ACR (Line 212-213) states that it's calculated for only the correctly classified samples. Moreover, the highest improvement for SmoothMix is in cases with $\sigma=1$, but that leads to a large drop in testing accuracy. I understand an inherent trade-off between robustness and performance, but this trade-off seems more prominent for SmoothMix than the Gaussian baseline.
> >
> > **Re Q2.** The authors mention that SmoothMix can boost previous methods when the task is more challenging, e.g., when higher  $\sigma$ is used. The underlying principle behind adversarial examples is making infinitesimal perturbations to a sample. Does using higher $\sigma$ break this adversarial criterion?

---

> > > ### Author Response · Authors · 2021-08-14
> > > **Discussion response to Reviewer cLLo**
> > >
> > > Many thanks for providing feedback on your remaining concerns in time for this discussion period. We address your questions one-by-one in what follows.
> > >
> > > ---
> > > **Q1-1. “ACR is calculated for only the correctly classified samples?”**
> > >
> > > It is important to note that the *incorrectly* classified samples also affect ACR, in a way to decrease its value. From the definition in Line 214, one can check that ACR normalizes the sum of certified radius (of correctly-classified samples) by $|\mathcal{D}\_{\tt test}|$, not by the size of the correct test samples: i.e., the numerator of ACR implicitly accounts for the incorrect samples by “certified radius of 0”. This is also a key reason why we think ACR can be a reasonable metric to evaluate the trade-off between accuracy and robustness, and has guided researchers to develop better methods in the literature of certified robustness (i.e., it is highly non-trivial to find a method to achieve state-of-the-art ACR under the tested benchmarks). We will clarify this in the final draft.
> > >
> > > ---
> > > **Q1-2. “The trade-off seems more prominent for SmoothMix than the Gaussian baseline, e.g., in its large accuracy drop at $\sigma=1.0$?”**
> > >
> > > Nevertheless, as also mentioned in our previous response, we agree that ACR is just one of many possible metrics to quantify the trade-off, and some readers may feel that ACR does not fully capture the relationship depending on one’s preference in between accuracy and robustness. To better address your concern, we have additionally considered a simple variant of ACR as a new metric, namely $\gamma\text{-}\mathrm{ACR}:=\mathbb{E}\_{(x, y) \sim \mathcal{D}\_{\tt test}}[\gamma\text{-}\mathrm{CR}(x, y)]$, which sets $\gamma\text{-}\mathrm{CR}(x, y) := -\gamma$ if $\hat{f}(x) \neq y$ while $\mathrm{CR}(x, y)$ otherwise, i.e., giving more penalty for the incorrect samples. Here, $\gamma > 0$ controls the preference of clean accuracy over robust accuracy, where the case of $\gamma=0$ becomes equivalent to the standard ACR.
> > >
> > > As highlighted in the table below (which is taken from Table 1 at $\sigma=1.0$), our method of SmoothMix could still outperform all the baselines under this metric for a wide range of $\gamma = 0.5, 1.0, 2.0$. Observe that our model of $\eta =1.0$ could perform better (or worse) than $\eta = 5.0$ under $\gamma=2.0$ (or $\gamma=0.5,1.0$). This is because SmoothMix with $\eta =1.0$ makes less degradation in the clean accuracy than $\eta = 5.0$, offering a clearer comparison in certified accuracy with other baselines of competitive clean accuracies.
> > >
> > > | Method | ACR | $\gamma$-ACR (0.5) | $\gamma$-ACR (1.0) | $\gamma$-ACR (2.0) |  0.00 | 0.50 | 1.00 | 1.50 | 2.00 | 2.50 |
> > > |:----------|:----------:|:----------:|:----------:|:----------:|:----------:|:----------:|:----------:|:----------:|:----------:|:----------:|
> > > | Gaussian | 1.620 | 1.600 | 1.582 | 1.545 | **96.3** | 91.4 | 79.8 | 59.4 | 32.5 | 10.9 |
> > > | Stability | 1.634 | 1.619 | 1.601 | 1.566 | **96.5** | 91.6 | 80.7 | 60.5 | 33.4 | 11.2 |
> > > | SmoothAdv | 1.779 | 1.757 | 1.736 | 1.694 | **95.8** | 90.6 | 80.8 | 64.6 | 43.3 | 22.2 |
> > > | MACER | 1.598 | 1.556 | 1.514 | 1.430 | **91.6** | 83.5 | 71.1 | 55.7 | 38.4 | 20.0 |
> > > | Consistency | 1.740 | 1.713 | 1.688 | 1.637 | **95.0** | 89.7 | 79.7 | 63.6 | 41.7 | 20.3 |
> > > | **SmoothMix ($\eta = 1.0$)** | 1.788 | 1.768 | 1.746 | **1.702** | **95.5** | 90.5 | 80.6 | 64.3 | 43.2 | 23.9 |
> > > | **SmoothMix ($\eta = 5.0$)** | **1.820** | **1.790** | **1.758** | 1.695 | **93.7** | 88.1 | 77.9 | 62.7 | 44.8 | 28.9 |
> > >
> > > ---
> > > **Q2-1. “Does using higher $\sigma$ break the adversarial criterion of making infinitesimal perturbations?”**
> > >
> > > As shown in Table 1, the certified robustness that even "$\sigma=1.0$" smoothing could offer is currently at most $|| \delta || < 3.0 \sim 4.0$, which is still not-too-large to completely change the semantics for most natural images. We believe pursuing provable robustness for such “moderate”-sized perturbations would be an important first-step to extend the current notion of adversarial robustness to many recent attempts in *out-of-distribution generalization*, e.g., towards stronger (or more flexible) adversaries [Kang et al., 2019; Tramer et al., 2019], or corruption robustness [Hendrycks et al., 2020; 2021], just to name a few.
> > >
> > > ---
> > > - [Kang et al., 2019] Transfer of Adversarial Robustness Between Perturbation Types, 2019.
> > > - [Tramer et al, 2019] Adversarial Training and Robustness for Multiple Perturbations, NeurIPS 2019.
> > > - [Hendrycks et al., 2020] AugMix: A Simple Data Processing Method to Improve Robustness and Uncertainty, ICLR 2020.
> > > - [Hendrycks et al., 2021] The Many Faces of Robustness: A Critical Analysis of Out-of-Distribution Generalization, ICCV 2021.

---

### Author Response · Authors · 2021-08-10
**Common response to all the reviewers**

Dear reviewers,

We thank all the reviewers’ efforts to improve our manuscript. This common response addresses concerns raised by multiple reviewers, viz., on the significance of our empirical results.

---

**Q. The improvements seem not significant.**

At some angle, the empirical gains in Table 1 and 2 may look seemingly small, especially when viewed in the average certified radius (ACR). Nevertheless, we still believe that our improvements can be useful to the NeurIPS community, considering that improving the frontier of ACR has been challenging given the previous efforts, yet our method could offer a consistent gain upon the previous state-of-the-arts. We further detail this and highlight some aspects to emphasize the significance of our contributions in what follows:

- **Our improvements are consistent**: Considering our efforts on keeping the hyperparameter choices as simple as possible, it is an important aspect that our improvements are *consistent* across many axes (as also noted by Reviewer jNUT); Table 1, 2 and Table 4 (in Appendix E) confirm the consistency across datasets and the choice of $\sigma$, and the ablation study (Section 4.3) does across the choice of hyperparameters. We additionally note that Table 5 in Appendix F shows the consistency of our method across multiple seeds of training. We believe these results strongly support the promise of our new design of adversarial training motivated by the overconfidence of “distant” adversarial examples, and opens many interesting future questions to better understand the properties of smoothed classifiers.

- **Our improvements are more significant on challenging tasks**: What makes our results more interesting is that they show SmoothMix can boost previous methods more significantly when the task is more challenging, e.g., when higher $\sigma$ is used (i.e., $\sigma=1.0$ in Table 1, and $\sigma=0.5$ in Table 2; see the below table for a summary), or on ImageNet dataset, while keeping the best performance at relatively “saturated” cases, e.g., $\sigma=0.25$ in Table 1 and 2. For example, in case of ImageNet with $\sigma = 0.5$ as highlighted in the below table (see Appendix E for the details), SmoothMix could achieve a quite significant improvement compared to the previous efforts on the same setup: e.g., we improved the previous best result reported (“MACER”) by 0.015 (0.831 → 0.846), while MACER did on its previous one (“SmoothAdv”) by 0.006 (0.825 → 0.831).

| Dataset | $\sigma$ | Gaussian | Consistency | SmoothAdv | MACER | _SmoothMix_ |
|:--------:|:-----:|:-----:|:-----:|:-----:|:-----:|:-----:|
| MNIST | 1.0 | 1.620 | 1.740 | 1.779 | 1.598 | **1.823** |
| CIFAR-10 | 0.5 | 0.525 | 0.720 | 0.684 | 0.691 | **0.737** |
| ImageNet | 0.5 | 0.733 | 0.822 | 0.825 | 0.831 | **0.846** |

Ultimately, our work challenges to “relax” the common assumption of small-norm adversaries in the literature of adversarial robustness: while focusing only on small $\ell_p$ balls for adversarial training, e.g., an $\ell_2$-ball of radius 0.5, has been practical yet foundational benchmarks, recent works suggest that such restrictions could be a source of robustness over-fitting [Kang et al., 2019], which makes it hard to bridge the adversarial robustness to a more realistic notion of robustness [Tramer et al., 2019; Maini et al., 2020; Kim et al., 2020]. By focusing on smoothed classifiers, our method could successfully incorporate more “distant” adversarial examples in training to improve the certified robustness without assuming a hard restriction on the perturbation norms: we believe our results are an encouraging signal towards developing a more “generalizable” adversarial training scheme in the future.


Thanks for your consideration,

Authors

---
- [Kang et al., 2019] Transfer of Adversarial Robustness Between Perturbation Types, 2019.
- [Tramer et al, 2019] Adversarial Training and Robustness for Multiple Perturbations, NeurIPS 2019.
- [Maini et al., 2020] Adversarial Robustness Against the Union of Multiple Perturbation Models, ICML 2020.
- [Kim et al., 2020] Adversarial Self-Supervised Contrastive Learning, NeurIPS 2020.

---

### Author Response · Authors · 2021-08-18
**A gentle reminder**

Dear Reviewers,

Thank you for your time and efforts in reviewing our paper.

We kindly remind that we are more than one week into the discussion period. We believe that we sincerely and successfully address your concerns/questions/misunderstandings/suggestions, with the results of the supporting experiments.

If you have any further concerns or questions, please do not hesitate to let us know.

Thank you very much!

Authors

---

### Decision · Program_Chairs · 2021-09-27

**Decision:**

Accept (Poster)

**Comment:**

All reviewers praised the simplicity of the idea of combining MixUp and adversarial training to boost certified accuracy and its significance.
At the same time, reviewers questioned the significance of the performance boost in terms of metric used (ACR) and potentially different behaviors across datasets. Furthermore, presentation and writing were deemed improvable.

Authors were quite responsive during the rebuttal and provided additional explanations and experimental results that answered the major concerns raised by the reviewers.
Fixing the presentation as suggested is expected for the camera-ready version.